# SCALABLE EVALUATION OF LANGUAGE MODELS WITH GENERATED GAMES

## ABSTRACT

We present `gg-bench`, a collection of generated game environments designed to evaluate the reasoning capabilities of language models. `gg-bench` is synthetically generated by (1) using an LLM to write game descriptions in natural language, (2) using the same LLM to implement each game in code, and (3) training RL agents via self-play on the generated games. We evaluate models based on their winrate against these RL agents by prompting them with the game description, current board state, and a list of valid moves, after which models output the moves they wish to take. `gg-bench` is challenging: general-purpose LLMs (GPT-4o, Claude 3.7 Sonnet) achieve winrates of 7-9% on `gg-bench` using in-context learning, while reasoning models (o1, o3-mini, DeepSeek-R1) achieve average winrates of 31-36%. Additionally, because `gg-bench` is a data generating process rather than a static benchmark, new evaluation instances can be created at will. We release the generated games, data generation process, and evaluation code in order to support future modeling work and expansion of our benchmark.

## 1 INTRODUCTION

Early researchers in artificial intelligence had broad ambitions of building systems capable of performing at or above human levels across arbitrary tasks. Often credited with the creation of the field of artificial intelligence, John McCarthy conjectured in his 1955 Dartmouth Conference proposal that "every aspect of learning or any other feature of intelligence can in principle be so precisely described that a machine can be made to simulate it" (McCarthy et al., 1955). However, in the subsequent decades, AI research narrowed significantly, focusing on more specific problem domains like chess, rule-based expert systems like DENDRAL (Buchanan et al., 1969), and knowledge engineering efforts like Cyc (Lenat et al., 1990; Russell & Norvig, 2016).

Concerned that the field had strayed too far from its initial ambitions, Goertzel & Pennachin (2007) coined the term "artificial general intelligence" in the early 2000s and urged researchers to move beyond "narrow AI." While the definition and usage of this term have been hotly debated in both AI and psychology (Sternberg & Detterman, 1986; Legg et al., 2007; Gardner, 2011), in this work we follow Chollet (2019) and use *general intelligence* to refer to the ability of a system to generalize and act in unseen contexts and environments.

In recent years, large language models (LLMs) have emerged as a potential stepping stone toward artificial general intelligence, and their performance on a wide variety of popular benchmarks has drastically increased in recent years (Bubeck et al., 2023). However, a growing concern is that these gains might not reflect true advancements in their ability to generalize to new domains, but might instead simply be the result of training on larger and more relevant datasets (Chollet, 2019). In other words, many tasks that were previously viewed as tests of out-of-domain generalization have now been moved into the training distributions of our models. As a result, it is an open question whether today's models can adapt and generalize to novel tasks in a way that would satisfy our definition of a generally intelligent system.

In this paper, we propose a scalable approach for evaluating whether models can generalize to new domains, leveraging a key observation: LLMs are capable of generating complex tasks that they themselves are incapable of solving. Under this view, benchmarks are not static lists of questions but *data generating processes*, such that individual task instances can be regenerated at will. This

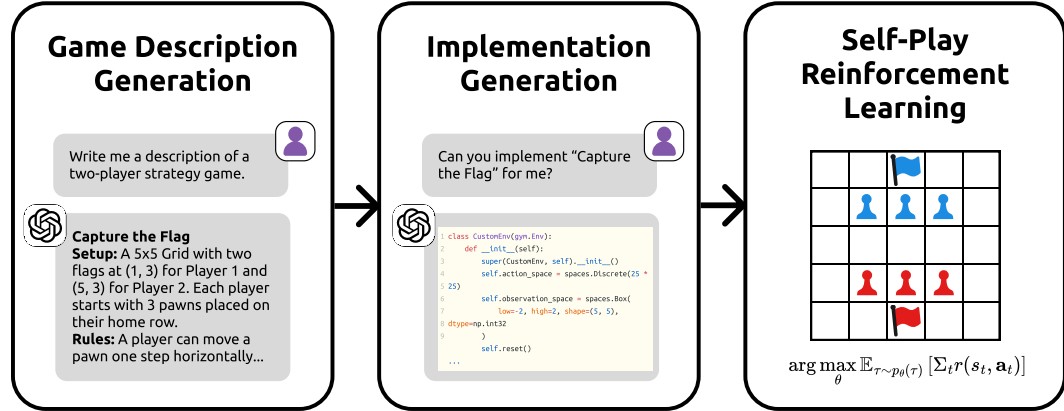

Figure 1: Overview of our benchmark creation process. We start by generating descriptions of two-player strategy games, after which we generate implementations of these games as Gym environments. Lastly, we employ self-play reinforcement learning to train agents on these games

approach allows us to generate new tasks in the result of data contamination, and also provides the possibility of generating more difficult tasks as stronger language models are developed and released.

We present `gg-bench`, a new benchmark consisting of games generated entirely by LLMs. The benchmark is created by first using LLMs to generate descriptions of two-player, turn-based strategy games designed to be played in a console environment. Then, using the same model, we generate Python implementations of each game in the form of Gym environments (Brockman et al., 2016). After this, we use self-play reinforcement learning to train agents on each of these games via proximal policy optimization (Schulman et al., 2017). Finally, in order to evaluate whether a target model can generalize to act in these generated games, we evaluate its winrate against the trained RL agents.

`gg-bench` is challenging: state-of-the-art LLMs such as GPT-4o and Claude 3.7 Sonnet achieve winrates between 7.5% and 9% on `gg-bench` using in-context learning, while reasoning models such as DeepSeek-R1 and o1 achieve average winrates between 31% and 36%. We analyze the diversity of generated games and identify common failure patterns of language models, revealing that their primary shortcomings are an inability to effectively strategize over multiple turns and to correctly generalize from game descriptions to new gameplay scenarios. Lastly, we release the dataset, code for generating the dataset and our experiments at `anonymous.tbd`.

## 2 GG-BENCH

The current iteration of `gg-bench` is a benchmark consisting of 126 datapoints, each of which is a two-player game. These 126 games are intended to be held fixed for fair and reproducible evaluation across models and papers; however, future releases of `gg-bench` may include additional or more challenging games. Each datapoint consists of the following components:

1. **Game description:** A natural language description of the game, describing its rules, objectives, and mechanics.

2. **Implementation:** A Gym environment implementation of the game. The gym environment consists of an action space, a `step` function, a `render` function, and a `valid_moves` function. An action space is a list of all possible actions that can be taken at any state, while the `step` function is used to apply an action to the current state of the game. The `render` function is used to convert the current state of the game to a string. The `valid_moves` function returns a list of valid moves given the current state of the game.

3. **Action space description:** A natural language description of each action in the action space. This is used to prompt the language model during evaluation.

The dataset is generated synthetically, with OpenAI o1 (OpenAI, 2024). An example of a generated game, code implementation, and action description can be found in Figure 2. Language models

```
# Number Duel

## Objective

Be the first player to **capture
all of your opponent's numbers**.
Utilize strategic selection and
timing to outmaneuver your opponent.
Victory is achieved when your
opponent has no numbers remaining
in their set.

## Setup

1. **Number Range Selection**:
   - Determine the value of **N**,
   the maximum number in each
   player's set. A recommended
   starting value is **N = 10**.

2. **Initial Number Sets**:
   - Each player receives a set of
   unique numbers ranging from **1
   to N** inclusive.
...

### Example Game Setup

- **N = 5**
- **Player 1's Numbers**: `{1, 2, 3,
4, 5}`
- **Player 2's Numbers**: `{1, 2, 3,
4, 5}`
- **First Attacker**: Player 1

### Round 1

- **Player 1** (Attacker) selects
**3**.
- **Player 2** (Defender) selects
**2**.
- **Reveal**:
  - Player 1: **3**
  - Player 2: **2**
- **Outcome**:
  - 3 (Attacker) > 2 (Defender):
  Attack successful.
  - **Player 2's number 2 is
  captured**.
  - Player 1's number **3 remains**
  in their set.
...
```

(a) An example game description from gg-bench. Parts of the description are elided with ... markers.

```python
class CustomEnv(gym.Env):
    def __init__(self, N=10):
        ...
        self.action_space =
        spaces.Discrete(N)
        self.observation_space =
        spaces.Box(
            low=0, high=1,
            shape=(2 * self.N +
            1,),
            dtype=np.float32
        )
        self.reset()
    def reset(self, seed=None):
        ...
    def step(self, action):
        ...
    def render(self):
        output = []
        output.append(
            f"Current role:
            {'Attacker' if
            self.current_role == 0
            else 'Defender'}"
        )
        ...
        return "\n".join(output)
    def valid_moves(self):
        ...
```

(b) Code for the Gym environment generated for the description provided. Implementation details are omitted and replaced with ... markers.

```
In the given gym environment for
the Number Duel game, the action
space indices range from 0 to N-1,
corresponding directly to the
available numbers a player can use
for their turn. Each index
represents a potential move, with
index i mapping to the number i+1
from a player's remaining set. For
example, choosing an action with
index 0 corresponds to selecting
the number 1, index 1 to selecting
the number 2, and so forth, up to
index N-1 for the number N. This
mapping allows players to choose
any available number for their
attack or defense from their
remaining numbers.
```

(c) Action description generated given the description and environment implementation.

Figure 2: An environment in gg-bench consists of three components: **(a)** a game description, **(b)** a Gym implementation, and **(c)** an action space description. Both the game description and action space description are available to the language model when prompted to select a move.

are evaluated based on their *winrates* against RL-based agents. In order to obtain high-quality and diverse games, we employ a multi-step generation and filtering process, outlined below:

| | Before Filtering | | | | After Filtering | | | |
|---|---|---|---|---|---|---|---|---|
| | Mean | Std | Min | Max | Mean | Std | Min | Max |
| Description length (tokens) | 1864.3 | 449.4 | 810 | 4505 | 1857.2 | 389.2 | 929 | 3158 |
| Code length (lines) | 126.6 | 41.7 | 54 | 408 | 125.5 | 39.7 | 61 | 255 |
| Action length (tokens) | 124.2 | 45.6 | 34 | 327 | 122.3 | 42.4 | 34 | 253 |
| Action space size | 78.6 | 584.6 | 2 | 13750 | 70.0 | 268.7 | 2 | 2500 |

Table 1: Basic data statistics for the 1000 games before filtering and the 126 games after filtering in `gg-bench`. "Action length" is the length of the natural language description of the action space.

## 2.1 GAME GENERATION

We start by prompting a model to generate 1000 unique two-player game rulebooks, each independently sampled. To ensure that language models can interact with the games, the prompt specifies that they must be playable in a console environment. We then generate implementations for each generated game in the form of a Gym environment (Brockman et al., 2016), along with a `valid_moves` function. Additionally, we generate descriptions mapping each action-space index to its corresponding in-game move. The cost for generating all games with o1 was $1162. The prompts used and implementation details can be found in Section C.

## 2.2 SELF-PLAY REINFORCEMENT LEARNING

We evaluate language models in terms of their winrates against RL-based agents. To obtain these agents, we employ proximal policy optimization (PPO) (Schulman et al., 2017). PPO works by optimizing a clipped surrogate objective, which constrains policy updates to prevent large changes, helping with stability.

We train agents using self-play reinforcement learning (Heinrich & Silver, 2016), where the PPO agent acts as both players in the generated environment. We train agents for $10^6$ timesteps and checkpoint every $2.5 \times 10^5$ timesteps. During training, at the start of each episode, we randomly sample a previously checkpointed agent to play against, except for the first $2.5 \times 10^5$ timesteps, where we play against a random agent. In addition, at each turn, we sample a random action with probability $\epsilon$, encouraging exploration. $\epsilon$ linearly decays from 1.0 to 0.1 over the training process. The agents are trained to maximize reward, which is 1 for a win, $-1$ for a loss, and 0 for a draw.

During inference, we employ Monte Carlo tree search (MCTS) to select actions. We sample 100 self-play trajectories starting at the current state using the trained RL agent, and log which trajectories result in a win for the current player. We then select the action at the root node leading to the child with the highest visit count, i.e., the action associated with the greatest number of simulated wins.

## 2.3 FILTERING

Throughout the generation process, we employ multiple filtering steps to ensure the quality of the generated games. These methods are outlined below:

**Keyword filtering.** Some games require large amounts of memory or computation, making it infeasible to train RL agents. For example, in word games, the action space is often exponential in the number of letters. To prevent this, we apply a regex and remove games with `**` in the action space.

**Execution filtering.** Some games have bugs in their implementations. We filter games by execution, checking whether the environment can be instantiated, returns the correct observation dimensions, and has a working render function. Game implementations are also generated with a function that returns a list of valid moves given the current state; for each environment, we play random agents against each other and filter games that throw exceptions even after taking moves from this list.

**Timeout filtering.** In initial experiments, we observed that win-condition checking and move application were often implemented incorrectly, resulting in never-ending games. To address this

problem, we implement timeout-based filtering by running an initial evaluation with GPT-4o-mini, where any games that take longer than 100 moves or over 1.5 hours to complete are filtered out. During this stage, we also filter out any games with an exception rate greater than 20%.

## 2.4 ESTABLISHING AN UPPER BOUND

We explicitly aim to demonstrate that the benchmark is *beatable*; that is, for each game included in `gg-bench`, there should exist some policy that is capable of consistently defeating the RL-based agent that we use to evaluate language models.

To empirically verify this, we consider RL agents checkpointed at four intervals throughout training. For each game, we evaluate every pairwise comparison of checkpointed agents across six matches. We then identify the pair of agents with the highest winrate disparity, ensuring one agent consistently outperforms the other. For `gg-bench`, we select the losing agent from this pair as the opponent that the language model must beat. Games lacking any agent pair with a winrate exceeding 80% are removed from consideration. Following this procedure, 126 distinct games remain. Among the remaining games, the winning RL agents achieve an average winrate of 91.02% against the chosen benchmark opponents, providing an existence proof that the games are practically beatable.

## 3 ANALYSIS OF GENERATED GAMES

We use o1 to generate natural language descriptions and code implementations for 1000 games; of these, 126 games passed all stages of filtering. We report basic statistics for these games in Table 1.

### 3.1 DIVERSITY OF GAMES

#### 3.1.1 EVALUATING CODE SIMILARITY

To measure the diversity and originality of the generated games, we employ DOLOS (Maertens et al., 2024), an open-source alternative to MOSS (Schleimer et al., 2003) for detecting code plagiarism. DOLOS assigns a similarity score in the range $[0, 1]$, where 0 indicates no detectable similarity and 1 an identical match. Across all game implementations, we observe a median maximum similarity score of $0.41$. For context, the example C and Java plagiarism datasets provided on the DOLOS website exhibit a median similarity score on the plagiarised documents of $0.72$. Additionally, we note that much of the similarity between game implementations is caused by boilerplate Gym code, e.g., having similar imports. The distribution of scores is shown in Figure 6 and additional statistics are presented in Table 6.

#### 3.1.2 WHAT TYPE OF GAMES ARE IN GG-BENCH?

To categorize the games in `gg-bench` by underlying strategy and core gameplay mechanics, we employed the goal-driven clustering method introduced by Wang et al. (2023). We use OpenAI o1 (OpenAI, 2024) to generate distinct categories for games such as number-based puzzles, grid-based movement games, and combinatorial strategy games. Then, we employ OpenAI o3-mini (OpenAI, 2025a) to assign each game to one of the proposed categories. Lastly, we group each of the categories into into five broader ones, described in Table 2. We provide the prompts used for categorization and the implementation details in Section E. We also provide more examples of games in Table 3.

Examining the distribution, we observe that number games, where the core mechanic involves choosing and manipulating numbers, often through arithmetic or number-theoretic reasoning, are the most common. We hypothesize this is due to number games being the easiest to implement and passing our filtering more than other games. Indeed, as shown in Figure 7, number games only make up 20.3% of the total game distribution prior to filtering as opposed to 36.7% post-filtering. We likewise see a consistent inclination toward random-chance mechanics and board games with clear action spaces, while combat-oriented games drop sharply—from 31.1% to 9.4% after filtering, likely because their win/lose state conditions are much more challenging to describe and implement.

To further verify that these findings generalize across different game types, we analyze winrate distributions across the five game clusters from Table 2. The relative performance ranking remains consistent across all clusters (Board, Number, Chance, Card, Combat), with reasoning-focused models

| Category | Share | Example | Core mechanics / objective |
|----------|-------|---------|----------------------------|
| Number | 36.7% | *Prime Claim* | Players alternately claim the integers 1–25. Primes add their own value; composites add their value *and* gift the factor-sum to the rival. Higher total after all picks wins; last pick breaks ties. |
| Board | 27.6% | *Isolation* | Players alternately claim unoccupied squares on a 13-square line that are *not* adjacent to any claimed square. The first to leave the opponent without a valid move wins. |
| Card | 14.6% | *High–Low Battle* | Players simultaneously reveal chosen cards 1–9 over five rounds, earning 1 pt for a higher card or 2 pts via the lower-previous-card tie-breaker. Highest total score wins. |
| Chance | 11.7% | *Digit Dilemma* | From a random 20-digit line, players alternately take one digit from either end and append it to their number; when the line is empty, the higher number wins (ties go to the second mover). |
| Combat | 9.4% | *Elemental Clash* | Two players start with 10 HP and four one-use spells. Elements interact rock-paper-scissors style; the winner deals damage, while ties hurt both. First to 0 HP—or with no spells left—loses. |

Table 2: Types of games present in `gg-bench` and illustrative examples from each category.

maintaining their advantage over instruction-tuned models; detailed results appear in Appendix G. Notably, the composition of wins differs markedly between model types. Instruction-tuned models derive a disproportionate share of their victories from chance-based games—with 13-17% of total wins concentrated in this single cluster—suggesting more specialized performance. In contrast, reasoning models exhibit substantially more balanced win distributions, achieving 23-44% of their wins across each cluster type, indicating more robust and generalizable game-playing capabilities.

## 3.2 FAITHFULNESS OF CODE IMPLEMENTATIONS

In order to measure the accuracy of the implementation of games, we manually evaluated a randomly selected subset of 50 out of the 126 filtered o1 games. Concretely, we annotated the descriptions, inspected the corresponding code, and then played through these generated environments ourselves. This verification step allowed us to directly assess whether each environment's implementation had faithfully matched the game mechanics described in the corresponding text. Of the 50 games we examined, all provided functional implementations. However, the implementation of number games sometimes provided hard-coded details. For instance, in *Divide and Conquer* (index 154), where players take turns dividing a shared number by some prime factor, we noticed that prime factors that can be used are hard-coded as a list, with all numbers $\leq 50$. While the game is still playable with this detail, it could error if the shared number is exceptionally high. However, we note that the language model is told (via the action description) that the list of primes is hard-coded.

## 4 EXPERIMENTS

**Models.** We evaluate various state-of-the-art LLMs: OpenAI ChatGPT (GPT-4o, GPT-4o-mini), Anthropic (Claude 3.7 Sonnet), Meta LLaMA (LLaMA-3.3-70B-Instruct). We also test reasoning models such as OpenAI o1, o3-mini and DeepSeek-R1. Small models (7/13B) are not tested due to the difficulty of the benchmark.

**Input format.** In order to get an action from a model, we prompt it with the game description, the current board state, a list of valid moves, and a description of what each move means. The model is then required to output a move from this list. If the model outputs a move not present in the list, we re-prompt the model and try again. The prompts used can be found in Section C.4.

(a) `gg-bench` winrates (mean ± 95% CI)

| Model | gg-bench |
|---|---|
| LLaMA-3.3-70B | 7.42 (± 2.78) |
| GPT-4o-mini | 7.64 (± 2.26) |
| GPT-4o | 8.94 (± 2.77) |
| Claude 3.7 Sonnet | 9.53 (± 3.05) |
| o3-mini | 31.08 (± 5.73) |
| DeepSeek-R1 | 32.50 (± 5.14) |
| o1 | **36.28 (± 5.95)** |

(b) Failure reasons for GPT-4o

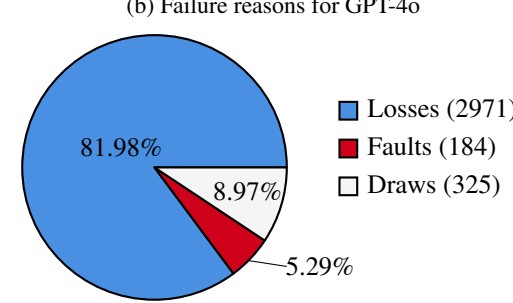

Losses (2971)
Faults (184)
Draws (325)

Figure 3: **(a)** Average winrates of various LLMs on `gg-bench` (30 games per matchup; 95% CIs in parentheses). **(b)** Breakdown of GPT-4o failures: "Faults" are invalid-move errors.

**Methods.** Each language model plays 30 games against an RL agent for every game in the benchmark. We calculate the winrate as the percentage of games the language model wins. The final score for each language model is the average winrate across all 126 games.

### 4.1 RESULTS

**Model performance.** As shown in Figure 3a, non-reasoning language models achieve relatively low winrates between 7% and 9%, while reasoning models achieve winrates between 31% and 36%. We observe that GPT-4o and Claude 3.7 Sonnet perform better than GPT-4o-mini and LLaMA-3.3-70B, indicating that larger models may have an advantage in handling the complexity of `gg-bench`. We also observe that reasoning models such as DeepSeek-R1 or OpenAI o3-mini achieve much stronger performance than non-reasoning models, suggesting that explicit reasoning capabilities are critical for success on `gg-bench`. This highlights the benchmark's emphasis on structured decision-making and long-horizon planning, which appear to benefit from models trained on reasoning tasks. For additional context, a random policy achieves only 5.36% winrate against our benchmark agents, while o3-mini achieves 70% winrate against the same random baseline (compared to 85.9% for our beatable RL checkpoint; see Section H). We report the cost and compute requirements of these experiments in Section A.

**Failure reason breakdown.** In Figure 3b, we show the distribution of failure reasons in `gg-bench`. The majority of losses are due to the RL agent winning, with a small percentage of draws and language model faults. The high percentage of RL agent wins suggests that current language models struggle with the strategic reasoning and adaptability required to succeed in these games. The low percentage of draws indicates that the games are well-designed and do not often result in stalemates.

**Example failed trajectory.** *Cross Over* (index 526) is a two-player strategy game where each side attempts to either invade the opponent's territory or eliminate all opposing pieces by moving along a linear track. On each turn, players can move each of their pieces either one or two steps along the track. In Figure 4, we show an example game where o1 (labeled LLM) loses to the RL agent. The early game is balanced until move 5, where the LLM moves piece P1-C to position 6, which the RL agent captures. After this, the LLM trades back and captures piece P2-B, but, in doing so, leaves its own backline undefended; notably, piece P1-A remains idle at position 0 for the entire game. This allows the RL agent to advance P2-C forward, and win the game. This trajectory illustrates the LLM's inability to evaluate long-term consequences of trades and territory exposure.

### 4.2 ARENA-STYLE EVALUATION BETWEEN LANGUAGE MODELS

In addition to evaluating language models against RL agents, we also run a small arena-style tournament in which models play directly against each other on `gg-bench`. We select five model GPT-4o, GPT-4o-mini, Claude 3.7 Sonnet, o3-mini, and o4-mini. Four of these models also appear in our main results, with o4-mini substituting for o1 as a comparable reasoning model. For each unordered model pair, we sample 10 games from `gg-bench`, and play 10 matches per game with

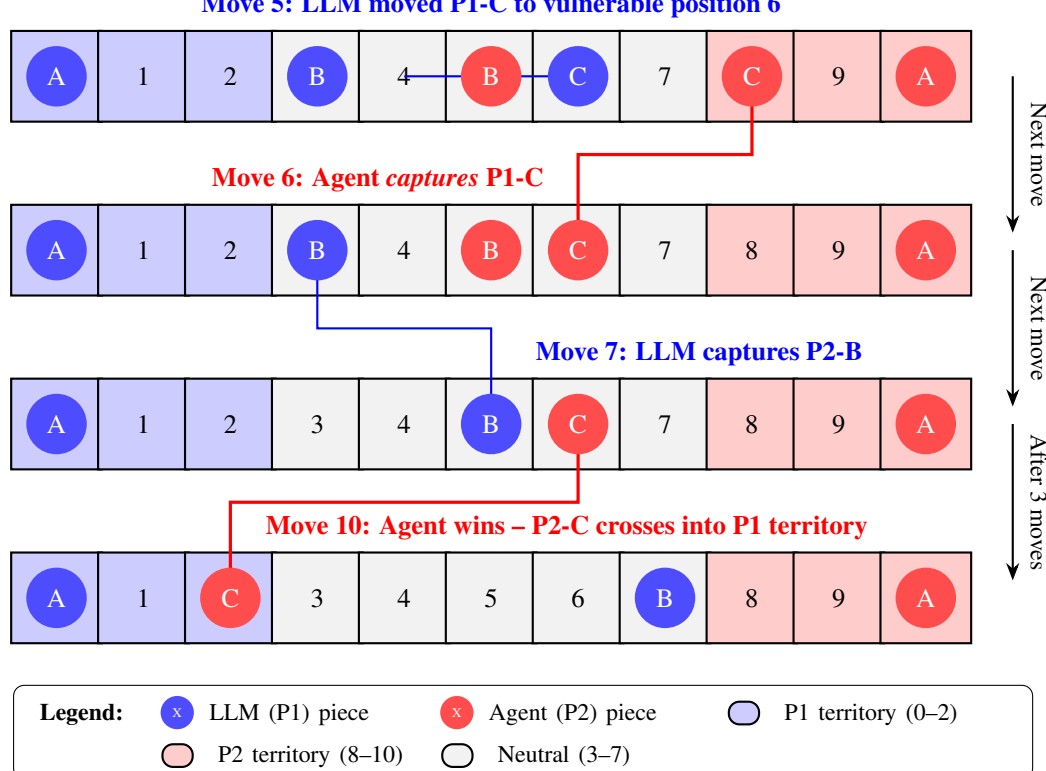

Figure 4: Example trajectory of *Cross Over* where o1 (labeled LLM) loses to the RL agent. Moves 0-4 are hidden as the game appears balanced until then, with both the LLM and the RL agent advancing their pieces forward. At Move 5, the LLM moves P1-C to position 6, highlighted by the blue arrow.

both models taking each side, yielding 100 games per pairing. We then fit Elo ratings by maximizing the likelihood of the empirical head-to-head score matrix (wins = 1, draws = 0.5, losses = 0), obtaining an order-invariant Elo-like rating for each model.

The resulting Elo scores, shown in Figure 5, mirror the pattern we see against RL agents in Figure 3a: reasoning-focused models dominate instruction-following models in head-to-head play.

We also provide the full $5 \times 5$ head-to-head matrix in Figure 5 and analyze its relationship to the aggregated `gg-bench` winrates. The matrix reveals several qualitative patterns that are not visible from Elo scores alone. First, although `o3-mini` loses consistently to `o4-mini`, it tends to "farm" the weaker pool of instruction-following models (GPT-4o, GPT-4o-mini, Claude 3.7) slightly more efficiently, which inflates its overall Elo and helps explain why its rating remains close to `o4-mini`. Conversely, the instruction-following models struggle against each other in a largely symmetric fashion, where none clearly dominates within their group, yet they are uniformly overpowered by the reasoning-focused models. Finally, the matchup between `o3-mini` and `o4-mini` is relatively competitive compared to their games against the instruction-following models, supporting the interpretation that the reasoning models form a separate, stronger tier on `gg-bench`.

## 4.3 SCALABILITY

We anticipate that more advanced language models will be capable of generating harder games. To substantiate this claim, we conducted a small-scale experiment comparing the quality of games generated by GPT-4o and OpenAI o1. We re-ran the generation pipeline of `gg-bench` using GPT-4o to create descriptions, implementations and action descriptions. After applying the syntactic and semantic filters described in Section 2.3 followed by the RL-agent upper-bound check in Section 2.4, 126 of the 1000 o1 games remained, whereas only 10 of the 1000 GPT-4o games survived.

| Model | Arena Elo |
|-------|-----------|
| o4-mini | 1688.9 |
| o3-mini | 1627.0 |
| GPT-4o-mini | 1410.7 |
| Claude 3.7 Sonnet | 1408.9 |
| GPT-4o | 1364.5 |

|  | o4-mini | o3-mini | Claude 3.7 | GPT-4o | GPT-4o-mini |
|---|---|---|---|---|---|
| o4-mini |  | 66.2 | 82.1 | 86.6 | 77.1 |
| o3-mini | 33.8 |  | 80.0 | 81.2 | 83.6 |
| Claude 3.7 | 17.9 | 20.0 |  | 56.2 | 50.8 |
| GPT-4o | 13.4 | 18.8 | 43.8 |  | 42.5 |
| GPT-4o-mini | 22.9 | 16.4 | 49.2 | 57.5 |  |

Figure 5: We conducted an arena-style evaluation in which five models were paired against each other, with 100 games per pairing. **(Left)** Elo ratings for all models. **(b)** Head-to-head winrate matrix.

Manual inspection reveals a qualitative gap as well. 8 out of 10 of GPT-4o-generated games are near-identical variants of *Tic-Tac-Toe* (cf. Section F), whereas the o1 set contains a diverse collection of novel win conditions and action spaces. These findings provide preliminary evidence that model scale is proportional to the difficulty and quality of the games present in `gg-bench`. Consequently, this result suggests that `gg-bench` may be *future-proof*; any saturation of the benchmark can potentially be mitigated by re-running the pipeline with a better model.

## 5 RELATED WORK

**Benchmarking LLMs with games.** Games have long served as testbeds for measuring AI capabilities, leading to breakthroughs like Deep Blue for chess (Campbell et al., 2002), AlphaZero for Go (Silver et al., 2017), and Libratus for poker (Brown & Sandholm, 2018). Schaul et al. (2011) argue that games offer a scalable proxy for artificial general intelligence because they can be procedurally generated to span a broad spectrum of difficulties and skills. Recent work has begun to evaluate LLMs with games. Text-adventure suites such as Jericho (Hausknecht et al., 2019) are designed to test agents' abilities to parse narrative state and issue actions. GameBench (Costarelli et al., 2024) focuses on hand-picked environments (e.g. Battleship, Connect Four) chosen to stress distinct planning skills while avoiding games likely present in pre-training corpora. Topsakal et al. (2024) provide a leaderboard for grid-based game competitions. ZeroSumEval (Alyahya et al., 2025) conducts arena-style evaluations on LLMs in classic strategy games like chess and poker, as well as knowledge tests and persuasion games. VGBench (Zhang & Press, 2025) challenges vision-language agents to complete a suite of 20 commercially released Game Boy and MS-DOS titles, ranging from *Doom II* to *Pokémon Red*, using only raw pixels as input. Releases of both Claude 3.7 Sonnet (Anthropic, 2025b) and Gemini 2.5 Pro (DeepMind, 2025) emphasized the models' abilities to play Pokémon Red, citing it as a strong out-of-distribution test of strategic reasoning. In contrast to all these works, though, we focus on games which are also generated by language models.

**Scalable benchmarking.** Fixed test sets quickly saturate as models improve, prompting a shift toward *scalable* or partially synthetic benchmarks that continuously generate new tasks. BIG-bench (Srivastava et al., 2023) introduced a community-contributed suite of over 200 tasks covering logic, math, and common-sense reasoning, many of which are procedurally created to avoid memorization, with BIG-bench Hard (Suzgun et al., 2022) isolating the most challenging subsets. Dynabench (Kiela et al., 2021) uses a *dynamic adversarial* approach: humans interact with state-of-the-art models in the loop, crafting inputs that fool them; those failures are immediately added to the training and evaluation pool, preventing saturation and exposing model weaknesses in real time. SWE-bench (Jimenez et al., 2024) automatically generates test instances by extracting coding tasks from real-world GitHub issues. $\tau$-bench (Yao et al., 2024) follows a hybrid synthetic approach, combining manually designed schemas, LLM-generated dialogues, and human refinement to evaluate agent interactions with tools and users in realistic domains. In contemporary work, Absolute Zero (Zhao et al., 2025) uses LLMs to generate synthetic tasks which are used for training reasoning models. `gg-bench` inherits this

spirit of scalability: new games, code implementations, and RL agents can be regenerated on demand, reducing the potential risks of dataset contamination and benchmark saturation.

**Reasoning with language models.** Many recent advancements in language modeling have been driven by *reasoning*, or the use of additional inference-time compute in order to obtain higher-quality generations. Early work in this direction showed that prompting models to generate explicit step-by-step answers, i.e., a chain of thought, improved their arithmetic and logical consistency (Nye et al., 2021; Wei et al., 2023). Training models to generate longer chains of thought via reinforcement learning has supposedly resulted in models such as OpenAI's o-series models (OpenAI, 2024; 2025a;b), Google's Gemini 2.5 Pro (DeepMind, 2025), Claude 3.7 Sonnet with "extended thinking" mode (Anthropic, 2025a) and DeepSeek's R1 (DeepSeek-AI et al., 2025), which have massively outperformed traditional LLMs on a wide range of benchmarks. Meanwhile, program-aided reasoning systems like PAL have models emit code that is executed to obtain verifiable answers, pushing performance beyond pure text-only reasoning (Gao et al., 2023). Tool-use agents (e.g. ReAct, Reflexion) further integrate search, calculators, or external APIs into the reasoning loop, enabling models to plan, act, and reflect iteratively (Yao et al., 2023; Shinn et al., 2023). Despite these advances, LLMs remain fragile in long-horizon and stateful settings, as evidenced by their performance in `gg-bench`.

## 6 DISCUSSION & FUTURE WORK

In contrast to traditional static benchmarks, the synthetic nature of `gg-bench` offers additional flexibility for future researchers looking to expand this dataset. We outline some key benefits below:

**`gg-bench` is scalable.** Because `gg-bench` is a data generating process, new games can be continuously generated using the existing pipeline, allowing the benchmark to expand as needed and mitigating potential risks of data contamination. More importantly, as model capabilities improve and the current iteration of the benchmark becomes saturated, we anticipate that stronger models will also be able to generate increasingly difficult games. RL agents will also likely scale alongside new algorithms and techniques; however, in the future, if training RL agents becomes a bottleneck, language models could also be evaluated in arena-style competitions against each other (Chiang et al., 2024; Alyahya et al., 2025). We predict that this scalability will result in `gg-bench` having greater longevity than most benchmarks.

**Controllable evaluation.** The data generating process of `gg-bench` is interpretable by design and therefore easily modifiable. For example, if future researchers wish to focus on games with specific design elements, or to modify aspects of existing games, they can easily do so by modifying our prompts or intermediate game descriptions. Additionally, the difficulty of the benchmark can also be tuned by selecting weaker or stronger RL agent checkpoints to evaluate language models against.

**Diverse evaluation.** Many existing benchmarks evaluate language models using known tasks or games, such as chess. However, because these tasks are often well-represented online (e.g., the web contains millions of games of chess), language models can obtain good performance by simply memorizing task-specific behavior rather than learning to adapt and reason in general settings. In contrast, `gg-bench` uses language models to design new games which are intended to differ from existing games that are over-represented in training corpora. Future work could further analyze the originality of our games and measure model performance as a function of game novelty.

Of course, the framework presented in this paper cannot possibly capture all aspects of general intelligence. For instance, the social intelligence of language models (Sap et al., 2022) cannot be evaluated in the context of two-player, zero-sum games. Furthermore, the definition and even the utility of the concept of *intelligence* have been hotly debated (Sternberg & Detterman, 1986; Legg et al., 2007). However, we hope that `gg-bench`'s ability to measure model performance beyond human-curated tasks will provide a useful signal to researchers looking to better understand and quantify the domain-general capabilities of language models.

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

## A    COST ANALYSIS

Since each game in `gg-bench` requires interaction with an RL agent, evaluating API models can be expensive. For GPT-4o-mini, GPT-4o, o3-mini and o1 the API costs were \$6, \$101, \$258 and \$2547 respectively, while for Claude 3.7 Sonnet, the cost was \$118. DeepSeek-R1 was run on the `together.ai` API, which cost \$461. LLaMA-3.3-70B was run locally on 4xNVIDIA A6000 GPUs. On average, for non-reasoning models, input tokens make up 99.95% of the cost, as the output tokens consist of a single number, i.e., the move the model makes. For reasoning models, however, the split skewed towards output tokens, with just 19.07% of the cost going to input tokens.

## B    GAME DESCRIPTIONS

In Table 3, we provide more examples of games present in `gg-bench`. These ten games illustrate the diversity of gameplay mechanics, ranging from arithmetic-based challenges (*Divide and Conquer*) to spatial reasoning (*Light Out Duel*), hidden information (*Line Duel*), and combinatorial strategy (*Order Challenge*). Each game is two-player and turn based.

| Game | Core mechanics / objective |
|---|---|
| **Palindrome Duel** | Players add X or O to either end of a sequence, avoiding formation of palindromes (length $\geq 3$). Forming a palindrome loses; reaching 11 symbols without palindromes wins. |
| **Divide and Conquer** | Players take turns dividing a shared integer by a chosen prime factor, aiming to be the one to reduce it exactly to 1. |
| **Power Match** | Each round, players choose a base (1–9) and an exponent (1–9); the higher resulting power wins (ties favor Player 2). |
| **Line Duel** | Players secretly play power cards (1–5) on a number line from –5 to +5. The difference on each turn pushes a marker; reaching the opponent's endpoint wins. |
| **Clash of Powers** | Players each hold the powers 1,2,4,8,16 and play one per round. Higher number wins unless it is exactly double the opponent's, in which case the smaller wins. First to 3 round-wins takes the game. |
| **Reach 27** | Players alternately add a number from 1 to 9 to a running total, racing to be the one who hits exactly 27. Exceeding 27 on your turn results in an immediate loss. |
| **Number Clash** | Both players start at 10 HP and simultaneously play cards 1–9. Damage dealt equals the difference between cards (ties deal 1 HP to both). First to reduce the opponent to 0 HP wins. |
| **Order Challenge** | Players build strictly increasing sequences by picking unique numbers 1–9. On each turn, a player must pick a number larger than their previous pick; failure to move loses. |
| **Light Out Duel** | From a row of seven lights, players alternately switch off either one light or two adjacent lights. The player who flips off the last remaining light wins. |
| **Command Clash** | Players start with 5 Command Points and secretly choose each turn among Charge, Attack, Special Attack, or Shield. The goal is to reduce the opponent's CP to zero. |

Table 3: Examples of two-player, turn-based strategy games present in `gg-bench`. Each row summarizes the core mechanics and objectives of a distinct game.

## C  IMPLEMENTATION DETAILS

In this section, we provide implementation details, such as prompts used for generation and evaluation or hyperparameters used during RL training.

### C.1  GAME DESCRIPTION GENERATION

We used the following prompt for game description generation:

```
You are tasked with creating a rule book for a new two player turn-based
game designed to be played in a command-line interface. The game should
be easy and simple to code, with no draw mechanism and should end
quickly. Furthermore, the game should be designed such that a skilled
player should be able to consistently beat an unskilled player. Make
sure that the game is unique, and is NOT similar to existing games such
as Go, Nim, Tic-Tac-Toe or Chess. The rule book should cover the
following aspects:

Objective: Clearly define the primary goal of the game. Explain how
players can achieve victory and what constitutes a win or loss.

Setup: Describe the initial setup of the game, including the arrangement
of game elements, player positions, and any starting conditions.
Game Components: List and explain all components involved in the game,
such as pieces, tokens, boards, or cards. Provide details on their
appearance, functionality, and any unique attributes.
```

```
Turns: Outline the structure of a turn, including the order of actions,
what players can do during their turn, and how turns progress.

Rules and Mechanics: Detail the core rules and mechanics of the game.
This should include movement or action rules, special abilities,
interactions between game components, and any unique game mechanics.

Scoring: Explain how points or other forms of scoring are tracked and
how they contribute to winning the game.

Examples: Provide example scenarios and command-line interactions or
sample turns to illustrate how the rules are applied in practice.

Ensure that the rule book is clear, organized, and comprehensive,
providing all necessary information to players while allowing for
strategic depth and complexity.
```

## C.2 ENVIRONMENT GENERATION

In order to generate a gym environment from a game description, we used the prompt below, providing an example Tic-Tac-Toe environment. We replaced `<GameDescription>` with the game generated using Section C.1.

```
<GameDescription>

Given this description, write a gym environment that implements this
game. Use gymnasium's API to define the environment. The action_space of
the environment should be a Discrete space, use spaces.Discrete to
define the action_space. The observation_space should be a Box space,
use spaces. The reward should be 1 if the current player wins, and -10
if the current player has played a valid move. The environment should
internally manage automatically switching between each player, it should
be designed for self-play reinforcement learning.

The environment should have the following methods:
- `reset()`: Reset the environment to its initial state. Returns
observation, info (dict).
- `step(action)`: Take a step in the environment. Returns observation,
reward, done, info (dict).
- `render()`: Return a visual representation of the environment state as
a string.
- `valid_moves()`: Return a list of integers of valid moves as indices
of the action_space.

Here is an example of how to define the environment:
```python
import numpy as np
import gymnasium as gym
from gymnasium import spaces

class TicTacToeEnv(gym.Env):
    def __init__(self):
        super(TicTacToeEnv, self).__init__()

        # Define action and observation space
        self.action_space = spaces.Discrete(9)
        self.observation_space = spaces.Box(
            low=-1, high=1, shape=(9,), dtype=np.float32
        )

        # Initialize the board
        self.reset()
```

```python
    def reset(self, seed=None, options=None):
        super().reset(seed=seed)
        self.board = np.zeros(9, dtype=np.float32)
        self.current_player = 1
        self.done = False
        return self.board, {}  # Return observation and info

    def step(self, action):
        if self.board[action] != 0 or self.done:
            return (
                self.board,
                -10,
                True,
                False,
                {},
            )  # Observation, reward, terminated, truncated, info

        self.board[action] = self.current_player

        # Check for win
        win_combinations = [
            [0, 1, 2],
            [3, 4, 5],
            [6, 7, 8],  # Rows
            [0, 3, 6],
            [1, 4, 7],
            [2, 5, 8],  # Columns
            [0, 4, 8],
            [2, 4, 6],  # Diagonals
        ]

        for combo in win_combinations:
            if all(self.board[i] == self.current_player for i in combo):
                self.done = True
                return self.board, 1, True, False, {}

        # Check for draw
        if np.all(self.board != 0):
            self.done = True
            return self.board, 0, True, False, {}

        self.current_player *= -1
        return self.board, 0, False, False, {}

    def render(self):
        board_str = "-------------\n"
        for i in range(3):
            board_str += "|"
            for j in range(3):
                if self.board[i * 3 + j] == 1:
                    board_str += " X |"
                elif self.board[i * 3 + j] == -1:
                    board_str += " O |"
                else:
                    board_str += "   |"
            board_str += "\n-------------\n"
        return board_str

    def valid_moves(self):
        return [i for i in range(9) if self.board[i] == 0]
```

```
Call the environment `CustomEnv`. Do not include any code that creates
the gym environment or tests it. Make sure the environment is fully
functional, requires no modifications and adheres to the requirements
specified in the prompt. Do not include any placeholder functions or
TODOs in the code.
```

### C.3 GENERATION ACTION DESCRIPTIONS

For generating descriptions as to what each index in the action space corresponds to, we used the following prompt, formatting `<GameDescription>` with the generated game description, `<PythonCode>` with the implementation of the game.

```
Here is a description for a two-player game:
<GameDescription>

Now, here is some python code that defines a gym environment for this
game:
```python
<PythonCode>
```

Your task is to write a brief explanation for the mapping between the
action space indices and moves in the game. Be concise with your answer
and avoid redundancy. Respond immediately with the explanation. Do not
include any other text in your response.
```

### C.4 LANGUAGE MODEL EVALUATION

For having the language model play against our RL agents, we used the following system prompt, formatting `<GameDescription>` with the generated game description and `<MoveDescription>` with the generation action space description.

```
Here is a description for a two-player game:
<GameDescription>

You will be prompted with a board state and a list of legal moves for
the current play. Your task is to pick the best move from this list.
Here is a description for what each move represents:
<MoveDescription>
```

Then, for each turn, we inserted the following prompt, replacing `<BoardState>` with the rendered board and `<LegalMoves>` with the list of legal moves the language model is allowed to take.

```
<BoardState>
Legal moves: <LegalMoves>
Pick the best move from the list of legal moves. Respond with the number
you wish to play. Do not include any other text in your response.
```

### C.5 SELF-PLAY REINFORCEMENT LEARNING

Reinforcement learning agents are trained using proximal policy optimization (PPO) (Schulman et al., 2017), using the implementation present in Stable Baselines3 (Raffin et al., 2021). PPO optimizes a clipped surrogate objective:

$$L^{\text{CLIP}}(\theta) = \mathbb{E}_t \left[ \min \left( r_t(\theta) \hat{A}_t, \ \text{clip}(r_t(\theta), 1 - \epsilon, 1 + \epsilon) \hat{A}_t \right) \right]$$

where $r_t(\theta) = \frac{\pi_\theta(a_t|s_t)}{\pi_{\theta_{\text{old}}}(a_t|s_t)}$ is the probability ratio, and $\hat{A}_t$ is the estimated advantage. The clipping prevents large, destabilizing updates by keeping $r_t(\theta)$ close to 1.

**Advantage estimation**    We use generalized advantage estimation (GAE) to compute $\hat{A}_t$:

$$\hat{A}_t = \sum_{l=0}^{\infty} (\gamma\lambda)^l \delta_{t+l}, \quad \delta_t = r_t + \gamma V(s_{t+1}) - V(s_t)$$

where $\gamma$ is the discount factor and $\lambda$ is the GAE decay parameter.

**Training setup**    Agents are trained via self-play for $10^6$ timesteps, with checkpoints saved every $2.5 \times 10^5$ steps. Initially, agents play against a random policy. After the first checkpoint, opponents are sampled uniformly from past checkpoints. Exploration is encouraged using $\epsilon$-greedy action selection, with $\epsilon$ decaying linearly from 1.0 to 0.1.

In addition, during training, we apply a timeout wrapper to the environment. If the environment crosses 100 moves from either players, the game terminates with an error and is filtered out. This is done to account for any games that unintentionally crept through the filtering present in Section 2.3. We provide the hyperparameters used during training in Table 4.

**Architecture**    We employ a standardized multi-layer perceptron with 2 hidden layers of 64 units each. This architecture remains fixed across all valid games generated, with only the input and output dimensions varying to match each game's observation and action space.

| Hyperparameter | Value |
|---|---|
| Learning rate | 3e-4 |
| Discount factor ($\gamma$) | 0.99 |
| GAE lambda ($\lambda$) | 0.95 |
| Clip range ($\epsilon$) | 0.2 |
| Batch size | 64 |
| Rollout length | 2048 |

Table 4: Key PPO hyperparameters used during training.

**Inference via MCTS**    At inference time, we apply Monte Carlo tree search (MCTS) to pick the move taken by RL agents. At the current state, we start by simulating 100 self-play rollouts using the trained policy. These are done by sampling a random action continuously from the probability distribution outputted by the RL policy, applied to both players. Each self-play rollout terminates when an ending state is hit. For each node, we keep track of the number of visits. Let $N(s, a)$ be the number of visits to child $a$ at root state $s$. We select the action:

$$a^* = \arg\max_a N(s, a)$$

i.e., the move leading to the most simulated wins.

C.6    FILTERING STATISTICS

Table 5 summarizes the attrition at each major stage of our pipeline. Starting from 1,000 initially generated environments, the 3-stage filtering process (described in Section 2.3) retained 316 environments, and the final upper-bound filtering step yielded 126 environments suitable for evaluation.

The 3-stage filtering removes 684 environments (68.4%) due to issues including syntax errors, execution failures, timeout during self-play, or training instability. The upper-bound filter then removes an additional 190 environments (19.0%) where the trained PPO agent achieved $> 90\%$ win-rate.

| Pipeline stage | Rejected | Remaining |
|---|---|---|
| Initial generation | — | 1,000 |
| 3-stage filtering (aggregate) | 684 (68.4%) | 316 |
| Upper-bound filtering (PPO $> 90\%$) | 190 (19.0%) | 126 |

Table 5: Environment counts at each major filtering stage. The 3-stage filter combines syntax validation, timeout checking, and training stability assessment. Percentages are relative to the initial 1,000 generated environments.

## D  PLAGIARISM ANALYSIS

For each game file in `gg-bench`, we computed its *highest pairwise similarity* to all other files using DOLOS (Maertens et al., 2024). Figure 6 shows the distribution of these maxima, and Table 6 summarizes the key statistics.

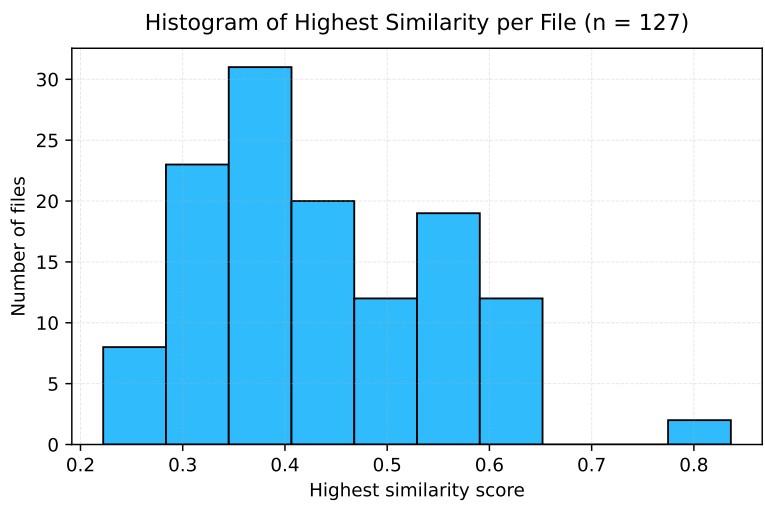

Figure 6: Distribution of the highest similarity score for every one of the 126 games in `gg-bench`.

| | Mean | Std | Min | 25% | 50% | 75% | Max |
|---|---|---|---|---|---|---|---|
| Highest similarity | 0.436 | 0.118 | 0.222 | 0.351 | 0.408 | 0.536 | 0.836 |

Table 6: Summary statistics of the highest similarity score observed for each game file ($n = 126$).

The median maximum-overlap score is $0.408$, and three-quarters of files fall below $0.54$, indicating only modest shared code beyond boiler-plate utilities. Only a few files exceed $0.70$ (the peak is $0.836$), and manual inspection attributes these cases to common helper functions rather than direct copying of gameplay logic. Overall, the analysis suggests that plagiarism within the corpus is limited and localised, supporting the benchmark's integrity.

## E  GOAL-DRIVEN CLUSTERING OF GAME DESCRIPTIONS

To analyze the diversity of environments in our benchmark, we applied a goal-driven clustering algorithm (PAS – Propose-Assign-Select) framework introduced by Wang *et al.* Wang et al. (2023) that provides interpretable, language-based explanations for each cluster. We defined our clustering goal as:

*"I want to cluster these game descriptions by game type, reflecting on their core themes and the primary strategy of the game."*

We ran the algorithm on a set of 126 game descriptions generated by our LLM pipeline. We used a powerful model (o1) to propose candidate cluster explanations and a smaller model (o3-mini) to assign texts to those explanations. The result of the assignment step is a binary matrix $A \in \{0,1\}^{N \times M}$, where $N = 126$ is the number of descriptions and $M$ is the number of candidate explanations. Entry $A_{i,j} = 1$ if description $i$ was judged to belong to cluster $j$, and 0 otherwise.

These assignments are then fed into an integer linear program (ILP) to select a compact set of clusters that covers each description at most once. Concretely:

- We introduce binary variables $s_j$ for each candidate cluster $j$, where $s_j = 1$ if cluster $j$ is selected.
- We introduce integer variables $m_i$ for each description $i$, enforcing

$$m_i = \sum_{j=1}^{M} A_{i,j}\, s_j, \quad 0 \le m_i \le 1,$$

  to ensure each description is covered at most once (forcing $m_i = 1$ if coverage is required).
- If a fixed number $K$ of clusters is desired, we add $\sum_{j=1}^{M} s_j = K$. Otherwise, we allow the solver to choose $K$.
- The objective minimizes the total number of uncovered descriptions.

$$\min \sum_{i=1}^{N} (1 - m_i) \; + \; \alpha \sum_{j=1}^{M} s_j \quad (\alpha = 0.5 \text{ by default}),$$

We solve this ILP using PuLP's CBC solver. The chosen clusters $j$ with $s_j = 1$ each form one final cluster, and descriptions $i$ with $A_{i,j} = 1$ are assigned accordingly.

The result are coherent groupings—e.g. number-based puzzles, grid-movement games, and combinatorial strategy games—while ensuring every description is placed exactly once.

### E.1 PROMPTING DETAILS

Our implementation is carried out entirely via three successively used prompts.

**Propose.** We first split the 126 descriptions into chunks. For every chunk, we query o1 the descriptions *in-context* as follows:

```
Below are a few examples of game descriptions:
{game_descriptions}
Goal: I want to cluster these game descriptions by game type, reflecting
on their core
themes and the primary strategy of the game. Please brainstorm a list of
{num_candidates} candidate explanations for clustering these texts. I
envision the following examples as valid themes: Card Game, Board Game,
Word Game, Abstract Strategy Game. Return the list as only numbered
items.
```

The model returns a simple numbered list and parsing those lines gives an initial pool of candidate clusters.

**Handling Duplicates.** The raw pool is concatenated and fed back to o1 with a meta-prompt

```
Here is a list of proposed cluster explanations:
{joined_explanations}
```

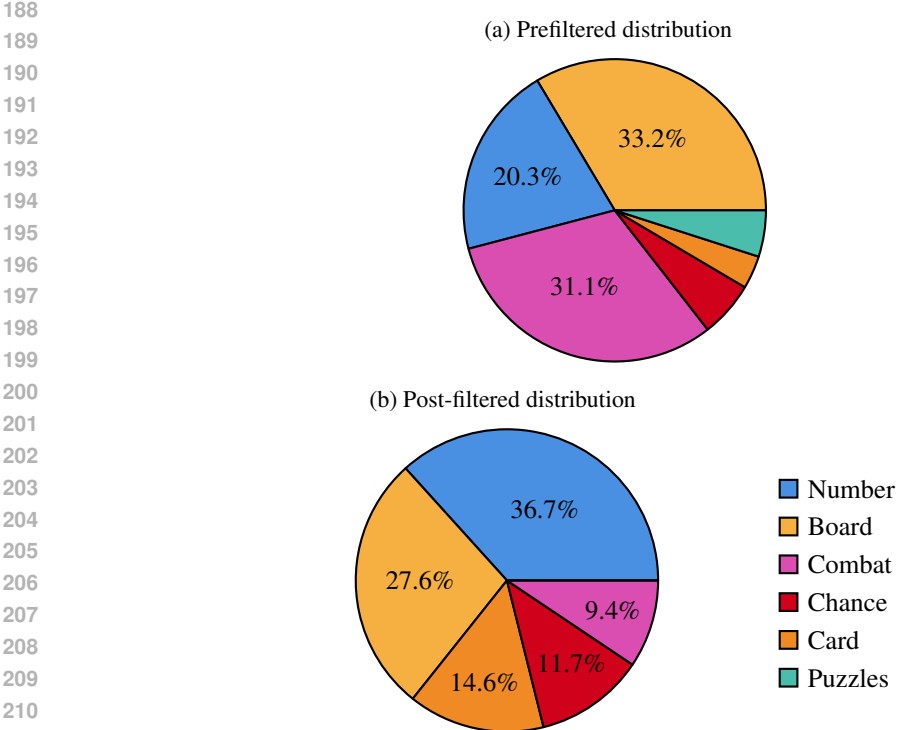

Figure 7: Genre-cluster distributions of o1-generated games **(a)** before and **(b)** after filtering. "Puzzles" is shorthand for "pattern puzzles."

```
Please remove any duplicates or near-duplicates, and remove any
explanation that is essentially a subset or redundant given another.
Then return the final list of unique, distinct cluster explanations as a
numbered list.
Do not add extra commentary.
```

This produces the final set of candidate explanations $\{e_1, \ldots, e_M\}$.

**Assign.** For every pair of (description $d_i$, explanation $e_j$ we query the assigner model (o3-mini) with

```
Cluster Explanation: {Example: Card Game: The game primarily involves
drawing, playing, or managing cards...}
Text: {Example: Game Title: Target Twenty-Three. Objective: Be the
player who reaches exactly 23...}
Question: Does the text belong to the cluster described above?
Answer with only either the 'Yes' or 'No' string and nothing else.
```

An answer of 'Yes' sets $A_{i,j} = 1$; 'No' sets $A_{i,j} = 0$. The resulting binary matrix $A$ is exactly the input to the ILP described above.

The pipeline helps keeps clusters concise, enforce disjoint cluster membership during the assignment phase, and preserves interpretability guarantees. We find that using reasoning models to do the task yields the highest quality explanation-based clusters.

## E.2 COMPARING DISTRIBUTION OF GAMES IN GG-BENCH PRE-FILTERING AND POST-FILTERING

**Clustering analysis**   As shown in Figure 7, we outline the game genre distributions for both the 1000 generated games, and the 126 that survive filtering. We notice three key changes when comparing the pre-filtering and post-filtering distributions:

- **Increase in card and number games:** Before filtering, "Combat" was the second-largest category at 31.3%, trailing only "Board" (33.2%). After filtering, "Number" games surge from 20.3% to 36.7%, overtaking "Board" and "Combat" as the largest category. Also noteworthy is the preference for card-based game mechanics, increasing from 3.5% to 13.3% after filtering.

- **Disappearance and shrinkage of niche clusters:** "Make-Sequence" or "Pattern Puzzle" games—where players must form exact patterns, such as in Color Bridge (which challenges two opponents to color exactly three adjacent nodes), or by arranging digits, symbols, and the like—are all but eliminated after filtering.

- **Relative stability of chance-based game mechanics:** After filtering, the "Chance" cluster climbs from 6.9% to 11.7%, about one in ten games, indicating that random-element mechanics remain appealing when backed by concrete descriptions and clear win conditions.

## F   SCALABILITY DETAILS

In Table 7, we provide summaries of the 10 GPT-4o games that survived filtering. We observe that 8 out of 10 games here are variants of or identical to Tic-Tac-Toe, where as the other two, *Numeral Clash* and *Sequence Duel* are both "running sum" games.

| Game | Core mechanics / objective |
|------|----------------------------|
| **Quantum Duel** | Players alternately place X/O on a $3 \times 3$ grid; first to form three in a row wins, otherwise the filled board resets the round. |
| **Dominion Duel** | Classic tic-tac-toe race on a $3 \times 3$ grid with no-draw rule—first three-in-a-row claims instant victory. |
| **Quantum Collapse** | Players drop X/O "energy fields" on a $3 \times 3$ matrix; aligning three triggers a "collapse" and wins the game. |
| **Cosmic Match** | Turn-based placement of X/O; first horizontal, vertical, or diagonal triple wins; no draws. |
| **Glyph Quest** | Place glyphs plus one-time Block, Swap, or Clear power; first to make three-in-a-row (or "V") wins. |
| **Quantum Clash** | Contest nodes on a $3 \times 3$ "circuit" using coin-flip challenges and energy tokens; win by a line of three activated nodes or total grid control within five rounds. |
| **Sequence Duel** | Players add $1-3$ to a shared running total; exact hit of target sum wins, overshoot loses. |
| **Elemental Duel** | Place/move tokens to claim Water (row), Fire (column), Earth (diagonal); first to hold all three patterns simultaneously wins. |
| **Quantum Flip** | Standard $3 \times 3$ alignment plus a one-use "flip" that converts an opponent's mark; forced resolution after five rounds; align three to win. |
| **Numeral Clash** | Draw numbers $1-5$; keep or assign to opponent; first to hit exactly 21 wins, overshooting loses. |

Table 7: Summaries of the 10 GPT-4o games that survived filtering. Each row summarizes the core mechanics and objectives of a distinct game.

| Model ↓ / Cluster → | Board | Number | Chance | Card | Combat |
|---|---|---|---|---|---|
| LLaMA-3.3-70B | 4.5% | 6.1% | 17.1% | 5.0% | 12.7% |
| GPT-4o-mini | 4.3% | 8.9% | 13.3% | 5.5% | 8.8% |
| GPT-4o | 3.8% | 8.4% | 14.6% | 11.6% | 14.8% |
| Claude 3.7 Sonnet | 7.2% | 8.2% | 17.7% | 9.6% | 11.3% |
| o3-mini | 24.1% | 38.5% | 31.8% | 23.4% | 33.5% |
| DeepSeek-R1 | 22.5% | 37.7% | 31.9% | 31.5% | 44.1% |
| o1 | 30.9% | 44.0% | 35.5% | 25.5% | 39.7% |

Table 8: Winrates (%) on `gg-bench` stratified by game category. Each entry is the average winrate of a model on games from the corresponding cluster (Board, Number, Chance, Card, Combat).

| Metric | Value |
|---|---|
| random agent win-rate | 5.36% ($\pm$1.70) |
| gg-bench agent win-rate | 85.86% ($\pm$4.08) |
| *Outcome breakdown (all games)* | |
| Random wins | 194 (5.16%) |
| gg-bench agent wins | 3,231 (85.89%) |
| Draws | 337 (8.96%) |
| Total | 3,762 |

Table 9: Random policy baseline versus the beatable PPO checkpoint on `gg-bench`. The large gap confirms that each game admits a reliably exploitable policy and that the benchmark is far from trivial.

## G  ADDITIONAL RESULTS BY GAME CATEGORY

Table 8 shows that the relative ordering of models is stable across all game types: reasoning-focused models (o1, o3-mini, DeepSeek-R1) consistently outperform instruction-tuned models on Board, Number, Chance, Card, and Combat games alike. While non-reasoning models achieve their highest win rates on Chance games (e.g., claude-3.7-sonnet at 17.7%, llama3.3-70b at 17.1%), reasoning models demonstrate substantially stronger and more diverse performance across all clusters. The gap is particularly pronounced in Number and Combat games, where reasoning models achieve 37-44% and 33-44% of their total wins respectively, compared to just 6-9% and 9-15% for instruction-tuned models. This pattern suggests that reasoning capabilities provide consistent advantages across diverse strategic domains, rather than specialized performance on particular game mechanics.

## H  RANDOM POLICY BASELINE AND EXPLOITABILITY

To further validate that `gg-bench` games are non-trivial yet systematically exploitable, we compare our "beatable" PPO checkpoints (used as opponents in the main results) against a uniform random policy.

The random agent selects a legal move uniformly at each decision point in the same environment interface used by PPO. We evaluate this random policy against the weaker PPO checkpoint on all 126 environments, using the same evaluation protocol as in the main experiments. In total we obtain 3,762 games. As shown in Table 9, the random agent wins only $\approx 5\%$ of games, while the PPO checkpoint wins $\approx 86\%$. Note that the best non-reasoning model we evaluated on, Claude 3.7 Sonnet, performed 9.53% on the same PPO agents.

We also compare the random policy to a reasoning-tuned LLM. On a subset of 10 randomly sampled games, with 10 matches per game, `o3-mini` achieves a 70% win-rate against the same random policy (95% CI: 65–75%), compared to 85.9% (95% CI: 81.8–89.9%) for the PPO checkpoint. In other words, `o3-mini` crushes the random agent almost as convincingly as the RL policy, while the random agent barely troubles the checkpoint.

Because each environment has well-defined transition dynamics and rewards, such a large gap between random, PPO, and reasoning-tuned LLMs is unlikely to arise from chance or single-move tactics alone. Instead, it suggests that strong models must systematically execute multi-step plans to approach the PPO upper bound, supporting our interpretation of `gg-bench` as a test of strategic, long-horizon reasoning rather than merely exploiting PPO-specific quirks.

