# OpenReview forum: "Scalable Evaluation of Language Models with Generated Games"
_ICLR.cc/2026/Conference — Submitted to ICLR 2026_

### Official Review · Reviewer_eDCL · 2025-10-25

**Soundness:** 3
**Presentation:** 3
**Contribution:** 3
**Rating:** 6
**Confidence:** 3

**Summary:**

The paper introduces a new benchmark designed to test the reasoning and generalization abilities of large language models (LLMs) using synthetically generated two-player games. Instead of relying on static datasets, gg-bench is a data-generating process: an LLM creates novel game descriptions, writes corresponding Gym environments in Python, and self-trains RL agents to act as opponents. Models under evaluation then play against these agents, with win rate as the performance metric.

**Strengths:**

The paper is clear, well-organized, and readable, making a technically complex topic accessible to a broad research audience. It effectively addresses an important challenge in AI evaluation: how to measure generalization and reasoning in language models beyond static benchmarks.

**Weaknesses:**

- The evaluation framework depends on RL-trained agents as the sole measure of difficulty.

- The authors could have conducted human evaluations on a small, representative subset of games. Such human-in-the-loop assessment would help contextualize RL agent performance, reveal potentially flawed game designs, and provide a richer reference point for what constitutes robust reasoning and generalization.​

- While the paper reports mean win rates for both RL agents and LLMs, it would be helpful if the authors explicitly compared RL agent performance to theoretical or empirical maximum scores for each game, normalizing achievements and clarifying the difficulty landscape. As for evaluation details, the paper states that each language model plays 30 games against the RL agent for every game in the benchmark (i.e., 30 trials per game), with the main result reported as the average win rate across the 126 games. However, there is no indication that scores are explicitly normalized relative to RL maxima, and RL agent scores are not directly shown in comparison to maximum possible scores.

- The paper does not clearly indicate on which specific games or categories language models perform better or worse. This makes it difficult to analyze the types of strategies or reasoning that differentiate successful LLMs, and may obscure systematic model strengths or weaknesses across different gameplay mechanics.

- The current evaluation restricts LLMs to picking moves turn-by-turn. An alternative approach, having LLMs generate complete code for game-playing policies or strategies, could explore their capacity for high-level planning, algorithm synthesis, or generalization at the program level. This could provide deeper insights into their actual reasoning abilities and offer a richer evaluation setting.

**Questions:**

Which types of games or strategies do current LLMs perform best on, and what does this suggest about the strengths and weaknesses of modern language models?​

How might the evaluation change if LLMs were asked to synthesize complete agents or policy code, rather than only making step-by-step move selections?

---

> ### Author Response · Authors · 2025-12-03
> **Author response to Reviewer eDCL (1/2)**
>
> Thank you for the constructive and thoughtful review.
>
> > Weakness 1: Dependence on RL agents as sole difficulty measure and lack of human evaluation on a subset.
>
> We agree this is a limitation. RL agents are:
> A practical proxy for “there exists a strong policy” and “the game is non-trivial.”
> However, they are not equivalent to human difficulty judgments or theoretical complexity.
> We agree this would be highly informative.
>
> To help extend our evaluation beyond RL agents as the sole difficult measure, we added an “arena-style” evaluation where LLMs play directly against each other on gg-bench, with no RL agents in the loop. We ran a 5-model round-robin tournament (o3-mini, o4-mini, GPT-4o, GPT-4o-mini, Claude-3.7-Sonnet), with 10 randomly selected games played 10 times each per model match-up (all pairings, both sides). We then fit Elo ratings by maximizing the likelihood of the observed head-to-head win-rates (order-invariant, i.e., not dependent on update order):
>
> LM Arena Elo Ratings
>
> | Model               | Elo   |
> |---------------------|-------|
> | o4-mini             | 1688.9|
> | o3-mini             | 1627.0|
> | gpt-4o-mini         | 1410.7|
> | claude-3.7-sonnet   | 1408.9|
> | gpt-4o              | 1364.5|
>
> Head-to-Head Win-Rate Matrix (%)
> | Model ↓  /  Opponent → | o3-mini | o4-mini | claude-3.7-sonnet | gpt-4o | gpt-4o-mini |
> |------------------------|--------:|--------:|-------------------:|-------:|------------:|
> | o3-mini            |   —     | 33.8    | 80.0               | 81.2   | 83.6        |
> | o4-mini           | 66.2    |   —     | 82.1               | 86.6   | 77.1        |
> | claude-3.7-sonnet  | 20.0    | 17.9    |   —                | 56.2   | 50.8        |
> | gpt-4o             | 18.8    | 13.4    | 43.8               |   —    | 42.5        |
> | gpt-4o-mini        | 16.4    | 22.9    | 49.2               | 57.5   |   —         |
>
> The reasoning-tuned models (o4-mini, o3-mini) sit clearly at the top, while instruction-following models cluster together at substantially lower ratings. This is largely consistent with the main gg-bench result. See Figure 3. Note: o4-mini performs quite similarly to o1 for significantly cheaper, and the LLM arena results maintain this fact.
>
> > Weakness 3: Normalizing scores relative to RL maxima / showing RL agent scores.
>
> Standard benchmarks like GLUE, MATH, and MMLU do not normalize by theoretical maxima or noise ceilings. Moreover, our upper-bound RL agents achieve >91% average win rate against the checkpointed gg-bench RL agents (Section 4.1), so normalization is unlikely to alter results or interpretations.

---

> ### Author Response · Authors · 2025-12-03
> **Author response to Reviewer eDCL (2/2)**
>
> > Weakness 4: No breakdown of which game types models do better on.
>
> Thank you for this suggestion. We have added this breakdown in Appendix G (Table 8). Non-reasoning models perform best on chance games, while reasoning models show stronger and more diverse performance across other game types, particularly number and combat games.
>
>
> (Percentage of each model's total wins from each cluster type)
> | Model ↓  /  Cluster → |   Board |  Number |  Chance |    Card |  Combat |
> |------------------------|--------:|--------:|--------:|--------:|--------:|
> | llama3.3-70b           |    4.5% |    6.1% |   17.1% |    5.0% |   12.7% |
> | gpt-4o-mini            |    4.3% |    8.9% |   13.3% |    5.5% |    8.8% |
> | gpt-4o                 |    3.8% |    8.4% |   14.6% |   11.6% |   14.8% |
> | claude-3.7-sonnet      |    7.2% |    8.2% |   17.7% |    9.6% |   11.3% |
> | o3-mini                |   24.1% |   38.5% |   31.8% |   23.4% |   33.5% |
> | deepseek-reasoner      |   22.5% |   37.7% |   31.9% |   31.5% |   44.1% |
> | o1                     |   30.9% |   44.0% |   35.5% |   25.5% |   39.7% |
>
> > Weakness 5: Alternative evaluation where LLMs synthesize complete agents.
>
> We fully agree that this is an exciting direction. There are two natural variants:
> - Code-generation agent: Prompt the LLM to output a Python policy (or heuristic function) that plays the game, then evaluate that policy against the RL agent.
>
> - Meta-policy: Have the model output a strategy description that is then executed by a separate, smaller decision procedure.
>
> Our current choice to evaluate move-by-move was motivated by:
> - Comparability with standard “LLM as an agent” setups (stepwise decision making),
> - Avoiding the complexity of sandboxing and executing untrusted code for each game.

---

### Official Review · Reviewer_qibi · 2025-10-29

**Soundness:** 2
**Presentation:** 3
**Contribution:** 2
**Rating:** 2
**Confidence:** 4

**Summary:**

The paper introduces gg-bench, a scalable benchmark of LLM-generated, two-player, turn-based games intended to evaluate LLM reasoning capabilities via gameplay against RL agents. The core problem addressed is that fixed, human-curated test sets saturate and risk contamination, making it hard to assess true out-of-distribution generalization. gg-bench generates games by first generating a game description/rulebook, then generating the code for the game as a Gym environment, and finally a PPO-based self-play agent is trained as the baseline to beat. The benchmark includes 126 filtered games (from an initial 1000). Evaluations show non-reasoning models perform significantly worse (~9% win rate over RL agent) compared to reasoning models (~30%).

**Strengths:**

- The benchmark is interesting, and the generation pipeline is useful for evaluating LLMs.
- Scalable engineering effort is a plus.
- Can provide meaningful empirical signals, with proper in depth analysis.
-

**Weaknesses:**

- Low novelty
- No concrete analysis of reasoning, which is the main position of the paper. There is insufficient ablations, investigation, analysis of model performance on the benchmark in relation to reasoning capabilities, which is the main axis that the benchmark evaluates. It is unclear if these games need reasoning, or if solutions are often illogical or unreasonable.
- LLMs are zero-shotted on the environment and evaluated against a trained model. While this isn't inherently bad, the empirical evaluation misses ablating to what degree LLMs can perform well. It lacks depth of study.
    - For example, ablations on multiple attempts might be valuable. The LLM could play multiple rounds, and learn from each round. This may also extend to testing how it reasons over past performance, etc.
- Information assymetry between RL agent, which also uses MCTS at inference time. This blends with the above point.
- Arguably, a human baseline is missing.
- Inference compute cost limits public usage, with high API costs needed to evaluate models and provide analysis.

**Questions:**

I only ask that the authors review my weaknesses above and either clarify any issues or provide content to fill in identified gaps.

---

> ### Author Response · Authors · 2025-12-03
> **Author response to Reviewer qibi (1/3)**
>
> Thank you for the detailed critique and for pushing on the “reasoning” aspect.
>
> > Weakness 1: Low novelty / “just a benchmark.”
>
> We agree this is primarily a benchmarking and evaluation paper, not a new algorithm. However, we do believe that there is some novelty in the way we approach the benchmark. Particularly, our contributions lie in:
> Turning “LLMs can generate tasks they can’t solve” into an explicit benchmark-as-process pipeline (rulebooks → code → RL opponents → evaluation).
> Introducing an RL upper-bound criterion for automatically selecting non-trivial, solvable games.
> Providing evidence that reasoning-tuned models separate cleanly from standard LLMs on these tasks.
> We will reframe our claims more clearly around these contributions.
>
> > Weakness 2: “No concrete analysis of reasoning” / lack of deeper ablations.
> We appreciate this feedback and have added concrete evidence that gg-bench games require multi-step strategic reasoning. Specifically, we introduce a random policy baseline to quantify task difficulty and isolate the contribution of reasoning capabilities. See Appendix G for the full breakdown and analysis.
> Setup: We evaluate a uniform random policy (selecting legal moves uniformly at random) against both our PPO checkpoints and reasoning models using identical evaluation protocols (3,762 games across all 126 environments for PPO comparisons; 100 games for o3-mini).
>
> | Opponent: Random Policy                | Win-rate | 95% CI        |
> |----------------------|----------|---------------|
> | o3-mini              | 70%      | 65–75%        |
> | Beatable-RL ckpt     | 85.9%    | 81.8–89.9%    |
>
> Analysis: The performance gaps provide concrete evidence that gg-bench games require strategic reasoning rather than simple tactics. Random policies win just over 5% against our checkpoint agents, establishing that these games are non-trivial and not solvable by chance alone. Notably, o3-mini performs substantially better against the random policy (70% win-rate), which aligns with findings from our LLM arena evaluation (Section 4.2) where reasoning models successfully beat LLM agents that are only slightly better than random: the strongest such agent, Claude 3.7 Sonnet, achieved just under 10% against gg-bench checkpoints. However, our checkpoint agents consistently outperform even the best reasoning models, maintaining an 86% win-rate against random opponents compared to o3-mini's 70%. This pattern reveals a critical distinction: while o3-mini can effectively exploit naive tactics and weakly strategic opponents, it fails to match the checkpoint agents' sophisticated multi-step planning. The gap between o3-mini's strong performance against random/weak policies and its substantially lower performance against our checkpoints demonstrates that gg-bench specifically tests the ability to execute long-horizon strategic plans rather than merely capitalizing on obvious mistakes or single-move opportunities.
>
> We have also supplemented this analysis by adding results on what models do better per game type. We have added this breakdown in Appendix G (Table 8). Non-reasoning models perform best on chance games, while reasoning models show stronger and more diverse performance across other game types, particularly number and combat games.

---

> > ### Author Response · Authors · 2025-12-03
> > **Author response to Reviewer qibi (2/3)**
> >
> > > Weakness 3: Information asymmetry (RL agent uses MCTS at inference).
> >
> > To check that our conclusions do not hinge on PPO-trained opponents, we added an “arena-style” evaluation where LLMs play directly against each other on gg-bench, with no RL agents in the loop. We ran a 5-model round-robin tournament (o3-mini, o4-mini, GPT-4o, GPT-4o-mini, Claude-3.7-Sonnet), with 10 randomly selected games played 10 times each per model match-up (all pairings, both sides). We then fit Elo ratings by maximizing the likelihood of the observed head-to-head win-rates (order-invariant, i.e., not dependent on update order):
> >
> > LM Arena Elo Ratings
> >
> > | Model               | Elo   |
> > |---------------------|-------|
> > | o4-mini             | 1688.9|
> > | o3-mini             | 1627.0|
> > | gpt-4o-mini         | 1410.7|
> > | claude-3.7-sonnet   | 1408.9|
> > | gpt-4o              | 1364.5|
> >
> > Head-to-Head Win-Rate Matrix (%)
> > | Model ↓  /  Opponent → | o3-mini | o4-mini | claude-3.7-sonnet | gpt-4o | gpt-4o-mini |
> > |------------------------|--------:|--------:|-------------------:|-------:|------------:|
> > | o3-mini            |   —     | 33.8    | 80.0               | 81.2   | 83.6        |
> > | o4-mini           | 66.2    |   —     | 82.1               | 86.6   | 77.1        |
> > | claude-3.7-sonnet  | 20.0    | 17.9    |   —                | 56.2   | 50.8        |
> > | gpt-4o             | 18.8    | 13.4    | 43.8               |   —    | 42.5        |
> > | gpt-4o-mini        | 16.4    | 22.9    | 49.2               | 57.5   |   —         |
> >
> > The reasoning-tuned models (o4-mini, o3-mini) sit clearly at the top, while instruction-following models cluster together at substantially lower ratings. This is largely consistent with the main gg-bench result. See Figure 3. Note: o4-mini performs quite similarly to o1 for significantly cheaper, and the LLM arena results maintain this fact.
> >
> > We also ran a small preliminary test on 10 randomly sampled games played 10 times each, enabling MCTS with rollout 5 on the LLM agent. With MCTS, o4-mini wins 23% (CI: 7.8–38.2%); without MCTS, it wins 40% (CI: 20.7–59.3%). Results are inconclusive due to low sample size, and we will provide a full ablation in the camera-ready.
> >
> > With MCTS
> > O4-MINI Win Rate: 23.00% std=23.26%
> > Confidence Interval: 7.80% - 38.20%, +- = 15.20%
> >
> > Without MCTS
> > O4-MINI Win Rate: 40.00% std=27.89%
> > Confidence Interval: 20.67% - 59.33%, +- = 19.33%
> >
> > > Weakness 4: Human baseline
> > We agree that establishing a measure of human performance would be an interesting avenue for future work, but we deemed it outside the scope of this paper due to the number and complexity of games that human annotators would need to learn.
> >
> > However, progress on this benchmark does not depend on a human baseline. For example, in a game like chess or Go, meaningful progress can be made even beyond the limits of human performance.

---

> ### Author Response · Authors · 2025-12-03
> **Author response to Reviewer qibi (3/3)**
>
> > Weakness 5: High inference cost
>
> We agree that generating games and training RL opponents is non-trivial in terms of API and compute costs. To clarify:
> Users of gg-bench do not need to regenerate games or retrain RL agents. Once the benchmark is released, most labs only need the fixed environments and pretrained RL agents, i.e., they only pay for their own model’s inference.
> The one-time cost is borne by us (and any future maintainer who wishes to regenerate a new version with a presumably newer frontier model). We will make this explicit and provide scripts that run evaluation solely with the already-trained agents.
> Traditional benchmarks rely heavily on human labor for data curation or costly data collection pipelines via API calls, etc gg-bench replaces this with a fully synthetic generation process. Although upfront generation has a cost (which is often tens of or hundreds of thousands of dollars), it scales much more favorably than ongoing annotation.
> We also note that modern reasoning-capable models are becoming substantially cheaper since the release of our initial paper. For example, recent OpenAI models have seen major reductions in per-token cost compared to earlier generations, especially for "reasoning" or "advanced" modes. For context, our initial gg-bench generation used o1 at 15 USD / 1M input and 60 USD / 1M output tokens, whereas newer models such as GPT-5 (5 USD/ 1M input, 15 USD / 1M output) and o3-mini (1.10 USD / 1M input, 4.40 USD / 1M output) are substantially cheaper, implying that future benchmark instantiations will be far less costly to produce.

---

### Official Review · Reviewer_b7uH · 2025-10-30

**Soundness:** 3
**Presentation:** 2
**Contribution:** 3
**Rating:** 6
**Confidence:** 3

**Summary:**

This paper introduces gg-bench, a novel benchmark designed to evaluate Large Language Models (LLMs) as reinforcement learning (RL) agents in diverse two-player games. The core idea is to leverage LLMs themselves to generate game descriptions and their corresponding Gym environment implementations, creating a scalable pipeline for task creation. To establish evaluation baselines, the authors train RL agents (using PPO with self-play) for each generated game. A crucial filtering step retains only those games where a trained RL agent can be clearly outperformed by a stronger version of itself (win rate >80%), ensuring the existence of a meaningful performance gap. The final benchmark consists of 126 such games. LLMs are then evaluated by their win rates against the "weaker" RL agent in each game. The results show that reasoning-capable models (e.g., OpenAI o1) significantly outperform standard ones, highlighting the importance of explicit reasoning for complex planning.

**Strengths:**

1) The use of RL upper-bound checks (PPO vs. PPO) for filtering is a major strength. It provides an objective criterion to ensure that each game in the benchmark is non-trivial yet solvable, and has a clear performance target, which enhances the reliability of the evaluation.

2) Evaluating LLMs against trained RL agents, rather than random or rule-based bots, creates a more challenging and informative testbed for strategic reasoning and planning abilities.

3) The significant performance gap between reasoning models (GPT-o1) and standard models provides strong empirical evidence for the value of chain-of-thought reasoning in complex decision-making tasks.

**Weaknesses:**

1) While the benchmark is theoretically scalable due to automated game generation, the practical scalability is limited. The need for manual intervention in feature engineering, state representation design, hyperparameter tuning, and debugging for each unique game makes the process of adding new tasks labor-intensive. The claim of "scalability" may therefore be more aspirational than fully realized in current practice.

2) The filtering mechanism favors games that are amenable to PPO-style deep RL. Games with sparse rewards, long horizons, or requiring symbolic reasoning might be filtered out even if they are excellent tests for LLMs, leading to a potential bias towards "RL-friendly" tasks.

3) The paper does not provide sufficient details on how the neural network architectures for the PPO agents are standardized or adapted across the vastly different game environments. This lack of transparency makes it difficult to assess the true level of automation and generalizability of the training process.

**Questions:**

1) Were there common failure modes during the PPO training phase? For instance, did many games fail the filtering step due to non-convergence, instability, or all agents performing at chance level?

2) The paper selects the "weaker" RL agent as the opponent. Have you experimented with evaluating LLMs against the stronger RL agent? If so, what were the win rates, and did they suffer from a "ceiling effect" as hypothesized?

---

> ### Author Response · Authors · 2025-12-03
> **Author response to Reviewer b7uH (1/2)**
>
> Thank you for the constructive review and for highlighting the RL upper-bound design.
>
> > Concern 1: Practical scalability limited by manual effort.
>
> We believe there is a misunderstanding here. The core contribution of our work is precisely that the pipeline is automatic. There is no per-game manual intervention, feature engineering, or hyperparameter tuning required. We use a single shared RL architecture and fixed hyperparameters (Table 4, Appendix C.5) across all 126 games with no game-specific modifications. The engineering effort was a one-time upfront cost to design the pipeline itself (prompts, filtering logic, training wrappers), after which generating 1000 games and filtering to 126 was fully automated. The only manual step was a one-time spot-check of 50 games to verify implementation faithfulness (Section 3.2), which validated the pipeline's quality rather than fixing individual games.
>
> > Concern 2: RL-friendly bias due to PPO filtering.
>
> To address the potential RL-friendly bias and demonstrate results that do not hinge on PPO-trained opponents, we added an “arena-style” evaluation where LLMs play directly against each other on gg-bench, with no RL agents in the loop. We ran a 5-model round-robin tournament (o3-mini, o4-mini, GPT-4o, GPT-4o-mini, Claude-3.7-Sonnet), with 10 randomly selected games played 10 times each per model match-up (all pairings, both sides). We then fit Elo ratings by maximizing the likelihood of the observed head-to-head win-rates (order-invariant, i.e., not dependent on update order):
>
> LM Arena Elo Ratings
>
> | Model               | Elo   |
> |---------------------|-------|
> | o4-mini             | 1688.9|
> | o3-mini             | 1627.0|
> | gpt-4o-mini         | 1410.7|
> | claude-3.7-sonnet   | 1408.9|
> | gpt-4o              | 1364.5|
>
> Head-to-Head Win-Rate Matrix (%)
> | Model ↓  /  Opponent → | o3-mini | o4-mini | claude-3.7-sonnet | gpt-4o | gpt-4o-mini |
> |------------------------|--------:|--------:|-------------------:|-------:|------------:|
> | o3-mini            |   —     | 33.8    | 80.0               | 81.2   | 83.6        |
> | o4-mini           | 66.2    |   —     | 82.1               | 86.6   | 77.1        |
> | claude-3.7-sonnet  | 20.0    | 17.9    |   —                | 56.2   | 50.8        |
> | gpt-4o             | 18.8    | 13.4    | 43.8               |   —    | 42.5        |
> | gpt-4o-mini        | 16.4    | 22.9    | 49.2               | 57.5   |   —         |
>
> The reasoning-tuned models (o4-mini, o3-mini) sit clearly at the top, while instruction-following models cluster together at substantially lower ratings. This is largely consistent with the main gg-bench result. See Figure 3. Note: o4-mini performs quite similarly to o1 for significantly cheaper, and the LLM arena results maintain this fact.
>
> We do agree that our filtering step could selects games with relatively dense and learnable reward signals under PPO + self-play. See Table 1, showing in short that filtered games remain diverse in complexity, with description lengths, code lengths, and action spaces spanning wide ranges similar to pre-filtering distributions.

---

> > ### Author Response · Authors · 2025-12-03
> > **Author response to Reviewer b7uH (2/2)**
> >
> > > Concern 3: Transparency: More details on PPO architectures and standardization across games.
> >
> > We have details about RL training, including algorithms and hyperparameters used during PPO in Appendix C.5 and Table 4. In short, the neural network architecture follows Stable Baselines3, where it is implemented as a standardized multi-layer perceptron with 2 hidden layers of 64 units each. This same architecture is applied across all 126 games, with the input/output dimensions automatically adapting to each game's observation/action space. More importantly, we have released the full benchmark of games, codebase, including all trained agents, environment implementations, and training scripts, allowing complete reproduction of our RL baselines and experiments.
> >
> > > Question 1: Common PPO training failure modes.
> >
> > Most (game, agent checkpoint) pairs fail the beatable-agent criterion (Section 2.4), filtering our dataset from 316 games to 126. Of the 190 games filtered out, PPO failed to learn meaningful strategies due to specific characteristics of the LLM-generated games: common failure modes included chance-level performance (all checkpoints at ~50%) and uniformly weak agents (all checkpoints <60% vs each other). This occurs when the LLM generates games with sparse reward signals or excessive stochasticity, preventing PPO from discovering learnable strategies within 10^6 training steps. Games that passed showed clear winrate disparities (averaging 91.2%) even though our imposed criterion was ‘only’ >80% between checkpoint pairs, demonstrating PPO successfully learned when the game design permitted it. See section E.2 also for pre-filtering and post-filtering game distributions.
> >
> > > Question 2: What happens if you evaluate against the stronger agent?
> >
> > We deliberately chose not to do this evaluation on the stronger agent and only on the weaker agent from each winning pair to ensure the benchmark remains a meaningful test rather than an impossibly hard task. The stronger RL agents in our checkpoints achieve an average winrate of 91.02% against the weaker agents we use as opponents. There is no guarantee that these agents are all beatable. Note, this is unlike the LLM arena experiment we showed since our benchmark is precisely able to beat them that makes the results meaningful. Note that this differs from the LLM arena experiment in Concern 2, where LLM models are also provably beatable by our benchmark. Our expectation (and informal checks) is that evaluating LLMs against the stronger RL agent would push winrates very close to zero for almost all models.

---

### Official Review · Reviewer_Jcg8 · 2025-10-30

**Soundness:** 3
**Presentation:** 4
**Contribution:** 3
**Rating:** 6
**Confidence:** 4

**Summary:**

The paper introduces gg-bench, a new scalable benchmark for evaluating the reasoning capabilities of LLMs. The core problem it addresses is the saturation and potential contamination of static benchmarks. Instead of a fixed set of tasks, gg-bench is a data generating process that uses an LLM to create new, unique two-player strategy games.

The process consists of three main stages:
- Game Generation: An LLM (here o1) is prompted to create natural language descriptions (rules, objectives, mechanics) for new games.
- Implementation: The same LLM is then prompted to implement these games as Python Gym environments.
- Agent Training: Reinforcement learning (RL) agents are trained via self-play (using PPO) on these generated games to create competent opponents.

LLMs are then evaluated by having them play against these trained RL agents. The model is given the game description, the current board state, and a list of valid moves, and must choose the best action. The primary metric is the winrate against the RL agent.
The key findings are that gg-bench is challenging and effectively differentiates between model classes. The paper provides a clear analysis of the generated games' diversity, code quality, and the strategic failures of LLMs.

**Strengths:**

- This benchmark is challenging and could serve as a testbed for very strong models, which is important nowadays since many evaluations get quickly saturated.
- The framework is inherently scalable. As language models become more capable, they can be used to generate more complex and difficult games, ensuring the benchmark's longevity and continued relevance.
- The paper is very well-written and easy to follow. The figures are clear and highly effective. Figure 1 provides a good overview of the pipeline, and Figure 4's breakdown of a failed game trajectory is helpful.

**Weaknesses:**

- The authors are transparent about the high cost of both generating the dataset ($1162 with o1) and running the evaluations ($2547 for o1). This presents a significant barrier to reproducibility and adoption for academic labs or independent researchers, potentially limiting the benchmark's widespread use.
- The paper notes that some implementations contained hard-coded details (for example a list of prime factors). This raises the question of how often the generated code might deviate from the natural language rules in ways that could be exploited by an RL agent but would be opaque to an LLM reading the rules. This could affect the fairness of the evaluation.

**Questions:**

- Please correct the typo in the title "Lanugage"
- Regarding the scalability, how would you scale 10x your benchmark? What would you target?
- Have you experimented with different RL algorithms or training regimes for the opponent agents? How can you be sure that the observed winrates reflect the LLMs' general strategic ability rather than their ability to exploit patterns specific to PPO agents?
- Do you have statistics on why games were filtered out? For instance, what percentage failed due to timeouts versus execution errors? This could illuminate the types of complex games that the current framework is systematically excluding.

---

> ### Author Response · Authors · 2025-12-03
> **Author response to Reviewer Jcg8 (1/2)**
>
> Thank you for the positive evaluation and detailed comments!
>
> > Concern 1: High cost is a barrier to adoption.
>
> We believe there may be a misunderstanding here. Our paper contributes both a generative process (the pipeline) and a concrete artifact (gg-bench-o1, the fixed set of 126 games with trained RL agents). While generating games and training RL opponents can be non-trivial, this is a one-time sunk cost that we have already absorbed. Users adopting gg-bench do not need to regenerate anything. They simply evaluate their models on our released environments with our pretrained agents, so the costs for all intents and purposes are negligible. The generation cost is explicitly not a barrier to adoption, only to creating future benchmark versions. We have added a clarification of this point in Lines 92-94.
>
> We also like to note that traditional benchmarks rely heavily on human labor for data curation or costly data collection pipelines via API calls, etc, gg-bench replaces this with a fully synthetic generation process. Even if future researchers regenerate the benchmark with newer frontier models, this one-time synthetic generation cost scales far more favorably than the ongoing human annotation required by traditional benchmarks.
>
> We also note that modern reasoning-capable models are becoming substantially cheaper since our original experiments. For example, recent OpenAI models have seen major reductions in per-token cost compared to earlier generations, especially for "reasoning" or "advanced" modes. For context, our initial gg-bench generation used o1 at 15 USD / 1M input and 60 USD / 1M output tokens, whereas newer models such as GPT-5 (5 USD / 1M input, 15 USD / 1M output) and o3-mini (1.10 USD / 1M input, 4.40 USD / 1M output) are substantially cheaper, implying that future benchmark instantiations will be far less costly to produce.
>
> > Concern 2: Faithfulness of implementations, hard-coded details.
>
> See section 3.2. On the fairness of evaluation, please refer to our points/experiments in response to Question 2 below (in the next comment).
>
> > Correct Title:
>
> Done. Thank you!
>
> > Question 1: Regarding the scalability, how would you scale 10x your benchmark? What would you target?
>
> Scaling gg-bench by 10x is straightforward because the entire benchmark is generated through a linear pipeline (game description, then code, then filtering, then RL training). See section Figure 1 and Section 2. The primary bottleneck is the pass rate of the generation/filters, not the evaluation itself.
>
> > Question 2: Have you experimented with different RL algorithms or training regimes for the opponent agents?
> We appreciate the question. We experimented with other RL algorithms - specifically REINFORCE and Deep Q-Learning, but found PPO to be the most stable and best performing.

---

> ### Author Response · Authors · 2025-12-03
> **Author response to Reviewer Jcg8 (2/2)**
>
> > Question 3:  Fairness of Evaluation: How can you be sure that the observed winrates reflect the LLMs' general strategic ability rather than their ability to exploit patterns specific to PPO agents?
>
> We have updated the paper. Please refer to section 4.2 and figure 5 for the full experiments and breakdown.
>
> In short, to check that our conclusions do not hinge on PPO-trained opponents, we added an “arena-style” evaluation where LLMs play directly against each other on gg-bench, with no RL agents in the loop. We ran a 5-model round-robin tournament (o3-mini, o4-mini, GPT-4o, GPT-4o-mini, Claude-3.7-Sonnet), with 10 randomly selected games played 10 times each per model match-up (all pairings, both sides). We then fit Elo ratings by maximizing the likelihood of the observed head-to-head win-rates (order-invariant, i.e., not dependent on update order):
>
> LM Arena Elo Ratings
>
> | Model               | Elo   |
> |---------------------|-------|
> | o4-mini             | 1688.9|
> | o3-mini             | 1627.0|
> | gpt-4o-mini         | 1410.7|
> | claude-3.7-sonnet   | 1408.9|
> | gpt-4o              | 1364.5|
>
> Head-to-Head Win-Rate Matrix (%)
> | Model ↓  /  Opponent → | o3-mini | o4-mini | claude-3.7-sonnet | gpt-4o | gpt-4o-mini |
> |------------------------|--------:|--------:|-------------------:|-------:|------------:|
> | o3-mini            |   —     | 33.8    | 80.0               | 81.2   | 83.6        |
> | o4-mini           | 66.2    |   —     | 82.1               | 86.6   | 77.1        |
> | claude-3.7-sonnet  | 20.0    | 17.9    |   —                | 56.2   | 50.8        |
> | gpt-4o             | 18.8    | 13.4    | 43.8               |   —    | 42.5        |
> | gpt-4o-mini        | 16.4    | 22.9    | 49.2               | 57.5   |   —         |
>
> The reasoning-tuned models (o4-mini, o3-mini) sit clearly at the top, while instruction-following models cluster together at substantially lower ratings. This is largely consistent with the main gg-bench result. See Figure 3. Note: o4-mini performs quite similarly to o1 for significantly cheaper, and the LLM arena results maintain this fact.
>
> On a separate note, our filtering mechanism is also designed to address this concern. As described in Section 2.4, every game in the final benchmark is required to satisfy an upper-bound check: a stronger PPO agent must reliably (≥80% win rate) defeat the weaker PPO agent trained on the same environment. This ensures that the weaker PPO agent has a consistent, exploitable strategy, and that the environment admits a clear performance gap between suboptimal and near-optimal play. In other words, for each (game, agent) pair, we know by construction that a better policy exists and is discoverable. The evaluation question is therefore not “Can the LLM defeat a perfect opponent?” or “Can the LLM exploit PPO artifacts?” but rather: Given access only to the natural-language rulebook and the current state, can the LLM discover strategies that approach the strong-PPO upper bound? Empirically, even state-of-the-art non-reasoning models fail to do so. They achieve near-random or slightly-above-random win rates (see experiment details below), despite the existence of a reliably exploitable policy. In contrast, reasoning-tuned models begin to close this gap, suggesting that the benchmark indeed captures differences in strategic reasoning rather than PPO-specific quirks. We view this alignment between known exploitability and LLM difficulty as evidence that gg-bench measures general strategic ability.
>
> > Question 4: Do you have statistics on why games were filtered out? For instance, what percentage failed due to timeouts versus execution errors? This could illuminate the types of complex games that the current framework is systematically excluding.
>
> This is a great suggestion. We added details in Appendix C.6.

---

### Official Review · Reviewer_y8Sy · 2025-10-31

**Soundness:** 2
**Presentation:** 2
**Contribution:** 1
**Rating:** 2
**Confidence:** 3

**Summary:**

The paper presents gg-bench,  which is a collection of generated game environments via LLMs. To be more specific, each game environment in gg-bench is generated by the following two steps. First, using an LLM to write game descriptions in natural language. Second, using the same LLM to implement each game in code. Afterwards, the reasoning capabilities of LLMs can be evaluated by playing them against RL agents that are trained via self-play on the generated games. Some experimental results are presented with regard to the winrates of several LLMs against the RL agents.

**Strengths:**

- using LLMs to generate game benchmarks is interesting.

- gg-bench is challenging to existing SOTA LLMs, as what has been shown in the experiments.

- the paper is very easy to follow.

**Weaknesses:**

- The technical contribution of the paper is to some extent insignificant. It is more of an interesting application of LLMs than a research paper. I would expect that some research questions have been answered, or some new knowledge has been generated.

- Since the game environments are generated using LLMs, it seems difficult to compare results across different papers, since different papers may use different LLMs. Even based on the same LLMs, the inherent randomness of LLM generation process may incur large variance to the evaluation process.

- gg-bench is only limited to two player zero-sum games.

**Questions:**

- Why gg-bench is so interesting or important that future LLMs should be tested on? What benefits/contributions does gg-bench bring to the exisitng community on the evaluation of LLMs reasoning abilities.

- How to ensure that results on gg-bench across different papers are comparable?

---

> ### Author Response · Authors · 2025-12-03
> **Author response to Reviewer y8Sy (1/2)**
>
> Thank you so much for reading our paper and providing your feedback! We address your concerns and questions below:
>
> > Concern 1: “Technical contribution is insignificant; more like an application than a research paper.”
>
> We respectfully disagree. Our work makes methodological contributions beyond a straightforward application: we introduce a non-static, self-generating benchmarking paradigm motivated throughout the paper. In particular, we leverage the “understanding–doing” gap to construct a pipeline that can continually regenerate new evaluation tasks. To our knowledge, this has not been done before.
>
> We introduce and open-source the entire synthetic generation pipeline that addresses the current pitfall of benchmarking capabilities of LLMs: benchmarking saturation and data contamination in light of an ever-expanding training set, particularly relevant for frontier models. While individual core components (LLM Code Generation, RL self-play) exist, their integration into a self-consistent evaluation framework to measure strategic long-horizon reasoning is novel. We also would like to point out that our games are synthetically generated, so measuring the “fairness” of the benchmark was important. We made the non-trivial design of ensuring every PPO-agent in the benchmark was repeatably beatable by another PPO-agent for every game.
>
> From experiments, we reveal that reasoning models (o1, o3-mini, DeepSeek-R1) achieve 31-36% winrates vs. 7-9% for standard LLMs.
>
> > Concern 2: Comparability Across Papers: “Since the game environments are generated using LLMs, it seems difficult to compare results across different papers…”
>
> We believe there may be a misunderstanding here. The current release of gg-bench consists of a fixed set of 126 games, and therefore, all results evaluated on this release are directly comparable across papers.
> To clarify: while gg-bench is designed as a generative process (which we view as a key contribution for future-proofing against saturation), we instantiate this process once, absorb the generation cost, and release the concrete dataset publicly. Any paper evaluating on gg-bench-o1 tests on the exact same 126 games with identical RL opponents, ensuring perfect reproducibility.
> The generative aspect addresses two separate benefits:
> 1. Future scalability: If the benchmark saturates, we can generate gg-bench-o2, gg-bench-deepseek, etc., with harder games as models improve (Section 4.2 demonstrates this with our GPT-4o comparison)
> 2. Contamination resistance: Unlike static benchmarks that may appear in training data, we can regenerate fresh evaluation instances if needed
>
> For the current release (gg-bench-o1):
> - Fixed seeds and filtering: Our pipeline uses deterministic filtering (Table 1) once seeds are set
> - Public hosting with versioning: We plan to maintain clear version history and leaderboards, similar to how benchmarks like MMLU or HumanEval operate
> - No cross-paper variance: Since all papers use the identical 126 games, there is zero variance from the generation process across evaluations
> The reviewer's concern about "inherent randomness of LLM generation" would only apply if researchers were regenerating new games for each paper, but this is explicitly not how gg-bench is used. We have added a clarification of this point in Lines 92-94.

---

> > ### Author Response · Authors · 2025-12-03
> > **Author response to Reviewer y8Sy (2/2)**
> >
> > > Concern 3: gg-bench is only a two-player zero-sum game.
> >
> > There are benefits to two-player zero-sum games. 1) they are easily measured in terms of win-rate, 2) allow for RL self-playing, and 3) test strategic reasoning without considering multi-agent coordination modeling. Two-player zero-sum games are also a well-established paradigm in both RL theory and for evaluating strategic reasoning, adopted by other benchmarks like ZeroSumEval (Alyahya et al., 2025) which tests LLMs on chess and poker, and GameBench (Costarelli et al., 2024) which includes games like Battleship and Connect Four among others. See section 5 on related work.
> > However, gg-bench offers key advantages over these existing approaches: unlike chess or poker (which have millions of training examples online), our synthetically generated games are contamination-resistant, with DOLOS median similarity of 0.41 compared to 0.72 for plagiarized code. Rather than 5-10 hand-picked games, we provide 126 diverse environments spanning number puzzles, board games, and combinatorial strategy, etc. See section 3.1.2. with demonstrated scalability to generate harder games as models improve (Section 4.2). Most importantly, our benchmark tests generalization to completely novel strategic contexts rather than memorization of established game theory. We note nonetheless that the expansion of games into multi-cooperative and multiplayer settings in general is very interesting, so we believe future work could heavily build on GG-bench ideas to do this!
> >
> > > Question: Why GG-bench is important for the community?
> > 1. Contamination Resistant: Unlike chess or Go, these games are unlikely to appear in training data. Our DOLOS analysis (median similarity 0.41 vs. 0.72 for plagiarized code) suggests substantial novelty.
> > 2. Scalable difficulty via self-regeneration: As models improve, we can generate harder games. This addresses the fundamental problem that static benchmarks saturate (e.g., MMLU, HumanEval are near-saturated).
> > 3. Reasoning gap: The 4x performance gap between reasoning and non-reasoning models demonstrates that gg-bench specifically stresses chain-of-thought capabilities in long-horizon planning.
> > 4. Interpretable failures: Failed trajectories (Figure 4) reveal specific strategic failures (inability to evaluate trades, leaving pieces idle), providing mechanistic insights into model limitations.

---

### Author Response · Authors · 2025-12-03
**Meta Response**

Hi! Here is a quick high-level summary of our changes:
- Multiple reviewers (Jcg8, b7uH, qibi, eDCL) raised potential concerns with evaluating LLMs against tabula rasa RL agents. To address these concerns, we also added an "LLM arena"-style evaluation in Section 4.2. We think this addition has made the paper significantly stronger! More details can be found in the PDF or in responses to individual reviewers.
- Reviewer eDCL asked for winrates broken down by game category, which we now provide in Table 8 in the appendix.
- Reviewer qibi asked for more ablations. We added random policy baselines and comparisons with o3-mini (a strong reasoning model) against our benchmarked agents in Table 9 in the appendix.

We also addressed a few misconceptions:
- It seems like a couple of reviewers (Jcg8, qibi) misunderstood the cost of generating the benchmark as the cost of evaluating new models on it. We edited the paper to clarify that this is a one-time cost borne by us and shouldn't affect adoption of the benchmark
- Reviewer y8Sy claimed that results on gg-bench wouldn't be comparable across papers, but this isn't true because the same set of games is intended to be used as the evaluation set across models and papers. We added a sentence in the paper to clarify this point.
- Reviewer b7uH claimed that our benchmark requires manual feature engineering, but this is simply not true. The entire pipeline for generating new games is automatic

---

### Meta-Review · Area_Chair_1yKc · 2026-01-07

**Summary:**

This paper introduces gg-bench, a benchmark for evaluating the reasoning abilities of language models using synthetically generated two-player games.
Reviewers generally agreed that benchmark saturation and data contamination are real problems and gg-bench is a attempt to address them. Several reviewers found the benchmark challenging and noted that it clearly distinguishes reasoning-tuned models from standard instruction-following models. At the same time, concerns were raised about the level of novelty, the reliance on PPO-trained agents as a proxy for difficulty, and whether the evaluation truly isolates reasoning ability rather than exploitation of RL agent behavior. Other concerns included limited ablations, lack of human baselines, unclear analysis of which game types drive performance differences.
The rebuttal and revisions partially addressed many of these issues through clarifications, additional experiments, and expanded analysis. While some reviewers remained unconvinced about the overall novelty and scope.

**Reviewer Concerns:**

### Concerns addressed by the rebuttal:

- The authors clarified that the released version of gg-bench consists of a fixed set of 126 games with pretrained RL agents, ensuring that results are directly comparable across papers.

- Misunderstandings about cost were addressed by clarifying that game generation and RL training are one-time costs borne by the authors, while users only need to run inference by their own.

- The authors added an arena-style evaluation in which LLMs play directly against each other. The resulting model rankings were consistent with the main benchmark results.

- Details about PPO architectures, hyperparameters, and training procedures were added, and the full benchmark (games, agents, and code) will be released to support reproducibility.

### Concerns that remain outstanding:

- Some reviewers continue to view the contribution primarily as a benchmarking effort rather than a methodological advance, and remain unconvinced that the novelty is sufficient.

- Difficulty is still largely defined via RL agent performance, and no human baseline is provided. This limitation is acknowledged but not resolved.

- The benchmark is restricted to two-player zero-sum games, and extensions to other game settings are left for future work.

- While clarified, inference cost for evaluating large models may still limit accessibility for some follow-up researchers.

**Reviewer Scores:**

Reviewer y8Sy: Initially 2. Concerns were not fully addressed;  score likely unchanged.

Reviewer qibi: Initially 2. concerns were partially addressed, score is likely to increase.

Reviewer Jcg8: Initially 6. Key concerns were addressed;  score likely unchanged.

Reviewer b7uH: Initially 6. Clarifications and additional evaluations address most concerns; score likely unchanged.

Reviewer eDCL: Initially 6. Added breakdowns and evaluations resolve major concerns; score likely unchanged.

---

### Decision · Program_Chairs · 2026-01-26

Reject